# In-situ observation of silk nanofibril assembly via graphene plasmonic infrared sensor

Chenchen Wu[1,2,3,9], Yu Duan[1,2,4,9], Lintao Yu [5,6], Yao Hu[7], Chenxi Zhao[5,6], Chunwang Ji[1,2], Xiangdong Guo[1,2,3,8], Shu Zhang[1,2,3], Xiaokang Dai[1,2,3], Puyi Ma[1,2,3], Qian Wang [7] ✉, Shengjie Ling [5,6] ✉, Xiaoxia Yang [1,2,3] ✉ & Qing Dai [1,2,3,8] ✉

Silk nanofibrils (SNFs), the fundamental building blocks of silk fibers, endow them with exceptional properties. However, the intricate mechanism governing SNF assembly, a process involving both protein conformational transitions and protein molecule conjunctions, remains elusive. This lack of understanding has hindered the development of artificial silk spinning techniques. In this study, we address this challenge by employing a graphene plasmonic infrared sensor in conjunction with multi-scale molecular dynamics (MD). This unique approach allows us to probe the secondary structure of nanoscale assembly intermediates (0.8–6.2 nm) and their morphological evolution. It also provides insights into the dynamics of silk fibroin (SF) over extended molecular timeframes. Our novel findings reveal that amorphous SFs undergo a conformational transition towards β-sheet-rich oligomers on graphene. These oligomers then connect to evolve into SNFs. These insights provide a comprehensive picture of SNF assembly, paving the way for advancements in biomimetic silk spinning.

Silk fibers, derived from natural proteins, possess remarkable properties, including outstanding mechanical strength, biocompatibility, and biodegradability[1,2]. These features make them highly attractive for diverse applications such as flexible brain–computer interfaces[3,4], tissue engineering[5,6], and wearable textiles[7,8]. Natural silk spinning involves a complex assembly process where silk fibroins (SFs) first form ordered silk nanofibrils (SNFs), which then organize into silk fibers with intricate hierarchical structures. While artificial spinning methods try to mimic natural conditions, the resulting regenerated silk often exhibits inferior mechanical properties compared to natural silk[9–12]. This disparity highlights the limitations in our current understanding of SNF assembly[13,14], hindering the development of biomimetic silk fibers.

Researchers have utilized diverse methods to explore the assembly process of SNFs. While simulation approaches have provided valuable insights into SNF structure and assembly, they face limitations on the accuracy due to the extended time scales and various length scales involved in the process[15–17]. For instance, high-resolution simulation methods, such as all-atom molecular dynamics (MD) simulations, demand significant computational resources, whereas simpler methods like coarse-grained MD simulations have low resolution[18]. These highlight the need for complementary experimental

[1]CAS Key Laboratory of Nanophotonic Materials and Devices, National Center for Nanoscience and Technology, Beijing 100190, China. [2]CAS Key Laboratory of Standardization and Measurement for Nanotechnology, National Center for Nanoscience and Technology, Beijing 100190, China. [3]Center of Materials Science and Optoelectronics Engineering, University of Chinese Academy of Sciences, Beijing 100049, China. [4]Henan Institute of Advanced Technology, Zhengzhou University, Zhengzhou 450001, China. [5]School of Physical Science and Technology, ShanghaiTech University, Shanghai 201210, China. [6]Shanghai Clinical Research and Trial Center, Shanghai 201210, China. [7]Department of Physics, University of Science and Technology of China, Hefei, Anhui 230026, China. [8]School of Materials Science and Engineering, Shanghai Jiao Tong University, Shanghai 200240, China. [9]These authors contributed equally: Chenchen Wu, Yu Duan. ✉e-mail: wqq@ustc.edu.cn; lingshj@shanghaitech.edu.cn; yangxx@nanoctr.cn; daiq@nanoctr.cn

characterization techniques to bridge the gap and achieve a comprehensive understanding of SNF assembly.

In experimental studies, researchers leverage a range of characterization techniques, including X-ray diffraction[19], scanning electron microscopy[15,16], transmission electron microscopy[17], and atomic force microscopy (AFM)[20]. These methods probe the micro/nanostructural properties of SNFs to deduce their assembly mechanism. Additionally, other techniques such as nuclear magnetic resonance[21,22], circular dichroism[23], small-angle X-ray scattering[19,24], fluorescence[25], Raman spectroscopy[26,27], sum-frequency generation spectroscopy[28,29] and Fourier-transform infrared spectroscopy (FTIR)[30], enable in-situ tracking of conformational changes during SNF assembly in solution. However, a major challenge lies in establishing a direct correlation between the conformational data obtained from spectroscopic techniques and the nanoscale morphologies revealed by microscopy methods. This hurdle arises because different techniques often require distinct sample preparation procedures, hindering the simultaneous investigation of both aspects on a single sample. Consequently, there is a growing interest in developing a method capable of simultaneously examining both conformational and morphological changes.

Graphene presents unique advantages for investigating SNF assembly. It acts as an assembly interface, immobilizing SNFs mainly through hydrophobic interactions and π–π stacking[31]. This characteristic not only accelerates SNF formation in SF solutions, reducing assembly duration from weeks to hours, but also aids in-situ observations of morphological changes during SNF assembly using AFM[32]. Moreover, graphene can be engineered with periodic nanostructures that enable the excitation of localized surface plasmons[33]. Graphene plasmon offer a significant enhancement for FTIR detection, achieving nanoscale sensitivity for detecting trace molecules[34–38]. Thus, the

attributes of graphene enable real-time monitoring of both secondary structure and morphological changes during SNF assembly, refining and validating various theoretical models for a deeper understanding of the assembly mechanism.

Here, we employ a graphene plasmonic infrared sensor in tandem with multi-scale molecular dynamics (MD) simulations to delve into the assembly process of SNF. By identifying secondary structure content and associating them with specific morphologies of different assembly intermediates, we reveal the key steps involved in SNF assembly. Subsequent experimental and simulation data underscore the pivotal roles of graphene interfaces and elevated temperatures in promoting the formation of elongated SNFs. These insights illuminate the SNF assembly process: SF molecules initially undergo a transition into β-sheet-enriched oligomers, then align and elongate to form SNFs, as demonstrated in Fig. 1a.

## Results

### Graphene plasmon-enhanced FTIR for probing SNFs

The assembly of regenerated SF in solutions serves as a widely used in vitro model for exploring the mechanism of silk spinning[39,40]. Thus, we utilized regenerated SFs extracted from *Bombyx mori* silkworm cocoon[41] to assemble on the graphene plasmonic infrared sensor (see Methods). The mean molecular weight of our utilized regenerated SFs is ~160 kDa, because the degumming process causes degradation of the SF peptide chains, as depicted in Fig. S1. To probe the assembly process of SNFs by FTIR, we utilized a graphene plasmonic infrared sensor (details in Fig. S2). The graphene nanoribbon width is ~60 nm and period-to-width ratio≈2:1 as shown in Fig. S3. The sensor was designed to a field-effect transistor based on monolayer graphene nanoribbons, as shown in Fig. 1b and Fig. S4. The gate voltage ($V_G$) was also applied for dynamically tuning the resonance frequency of

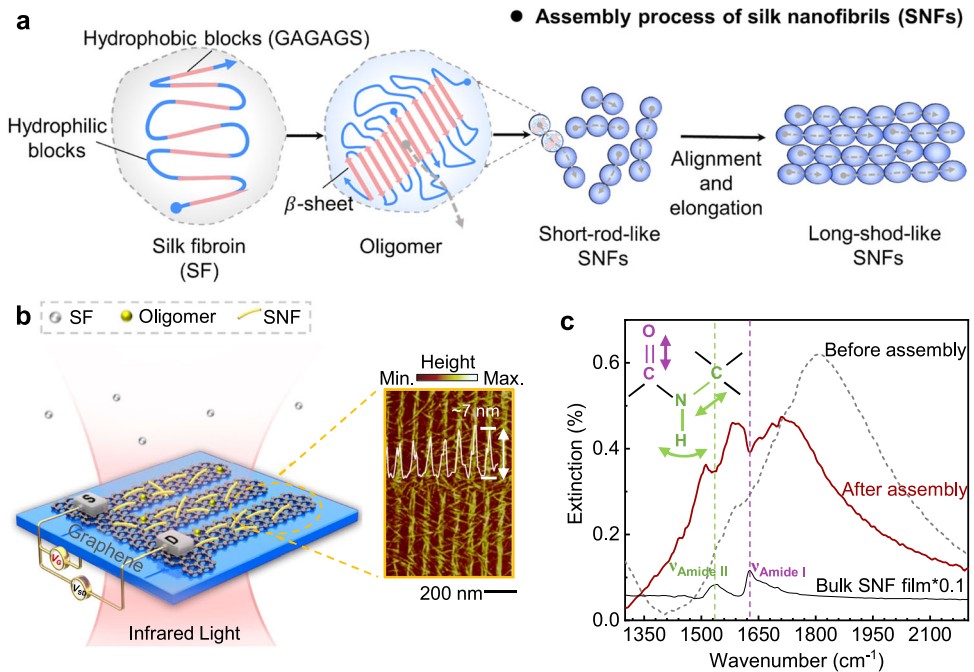

**Fig. 1 | SNF assembly and graphene plasmon-enhanced FTIR. a** Schematic diagram of speculative SNF assembly mechanism: Unfolded SF molecules first undergo a conformational change, forming β-sheet-rich oligomers; these oligomers then connect with nearby molecules and elongate into short-rod-like SNFs, with the β-sheet chains aligned along the elongation direction (indicated by gray dashed arrow). Finally, these short SNFs further align and elongate, becoming thicker and longer to form long-rod-like SNFs. **b** Schematic diagram of graphene

plasmon-enhanced FTIR for measuring SNF assembly (left panel) and a typical AFM of SNF adsorbed on the graphene nanoribbons (right panel). **c** The extinction spectra of graphene plasmon with (red curve) and without SNFs (black dashed curve). $T_{V_G}$ is measured when $V_G$ is −100 V, while $T_0$ is measured when $V_G$ is 100 V. The black curve is extinction spectrum of bulk SNF film which is reduced by 10 times for comparing. The assembly process occurs in an aqueous solution at 348 K, with a duration of 300 min.

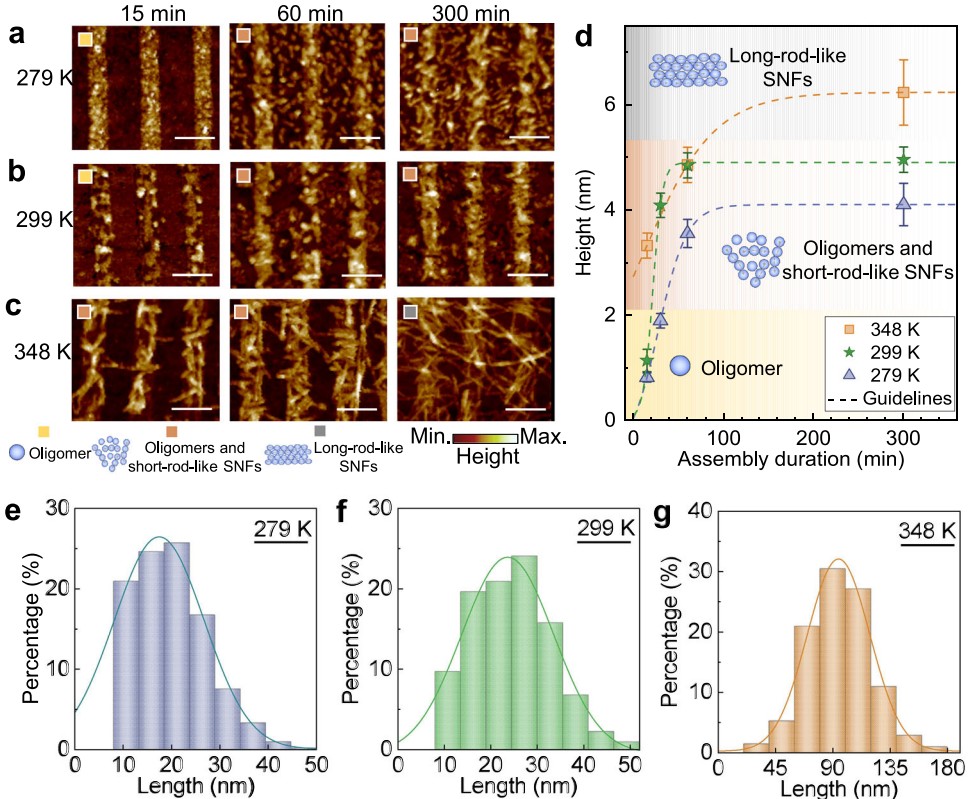

**Fig. 2 | Nanoscale morphology changes during SNF assembly.** Morphology of oligomers and SNFs adsorbed on graphene nanoribbons at (**a**) 279 K, (**b**) 299 K, and (**c**) 348 K. The scale bars are 150 nm. **d** The extracted height changes from (**a**–**c**). The data were collected from three different positions of graphene nanoribbons and expressed as mean values +/− SEM. The dashed curved are guidelines. The statistical length distribution of SNFs with a height of 3–4 nm at (**e**) 279 K, (**f**) 299 K, and (**g**) 348 K.

graphene plasmon to target molecular fingerprints (details in Note 1 and Fig. S5 of Supplementary information)[42,43].

In the right panel of Fig. 1b, a typical AFM image displays the SNFs selectively assembled on the graphene nanoribbons after 300 min of assembly. The total height of the assembled structure is approximately 7.0 nm, which includes ~5.0 nm-thick SNFs and ~2.0 nm-thick monolayer graphene. The observed height of the monolayer graphene exceeding 0.34 nm is attributed to poly(methyl methacrylate) (PMMA) residue contamination from the sensor fabrication[44]. The assembled SNFs are located on the edges and surface of the graphene nanoribbons (i.e., graphene plasmonic hotspots)[36], which enables graphene plasmonic infrared sensors to sensitively measure their chemical structure. The extinction spectra of the sensor were obtained by in-situ altering the gate voltage using the formula Extinction = $1 - T_{V_G}/T_0$, where $T_{V_G}$ represents the transmittance measured at a specific gate voltage ($V_G$), and $T_0$ is measured at the charge-neutral point ($V_{CNP}$) of graphene.

Figure 1c indicates two typical extinction spectra of the graphene plasmon infrared sensor: one before (black dashed curve) and the other after (red curve) SNF assembly. Upon SNF assembly, two prominent notch regions (i.e., 1538 cm$^{-1}$ and 1626 cm$^{-1}$) become evident on the graphene plasmon resonant peak. These dips align well with the peaks observed in the infrared spectrum of bulk SNF films (black curve), corresponding to the amide I band (i.e., purple dashed line) and amide II band (i.e., green dashed line) of as-assembled SNFs. These signatures appear as dips on the graphene plasmon resonance peak due to the destructive interference between the graphene plasmon and the SNF molecular vibrations. Graphene plasmon enable a robust infrared response from the nanoscale SNFs, facilitating the analysis of their secondary structures. Additionally, after the assembly of SNFs, the plasmon resonance of graphene nanoribbons experiences a

red-shift, whereas the same $V_G$ was maintained as before the assembly. This shift can be attributed to the negative charge carried by SFs (with an isoelectric point ~3.5) in the solution (pH ~6.8), resulting in the doping of graphene[32].

**Morphology analysis during SNF assembly**

We propose that the assembly process of SNFs involves nucleation, growth, and a final plateau. To reveal the intermediates at various assembly stages, we varied two key parameters: assembly duration (15 min, 60 min, and 300 min) and assembly temperature (279 K, 299 K, and 348 K). Adjusting temperature allows for control over the assembling rate[32]. Then the morphologies of the assembled SNFs on the graphene plasmonic infrared sensor were examined using AFM (details in Methods), as presented in Fig. 2a–c. It is worth noting that we immersed the same graphene plasmonic infrared sensors in SF solutions (10.1 μg/mL) to study the assembly process across different durations at each temperature.

At 279 K, oligomers with a height of ~0.8 nm appeared on the graphene surface after 15 min of assembly, as illustrated in Fig. 2a. These oligomers gradually evolve into short-rod-like SNFs with increasing assembly duration. After 60 min and 300 min, short-rod-like SNFs heights reach ~3.5 nm and ~4.1 nm, respectively. At 299 K, the initial assembly stage (15 min) shows similar behavior with the formation of oligomers, but with a slightly larger height of around 1.2 nm compared to those observed at 279 K, as shown in Fig. 2b. Subsequently, the morphology evolved from oligomers to dense short-rod-like SNFs with heights of ~4.8 nm and ~4.9 nm after 60 min and 300 min of assembly, respectively. At 348 K, the assembly process appears faster at this higher temperature. Even after only 15 min, short-rod-like SNFs with a height of about 3.3 nm are already observed, as depicted in Fig. 2c. As the assembly duration increased to 60 min and

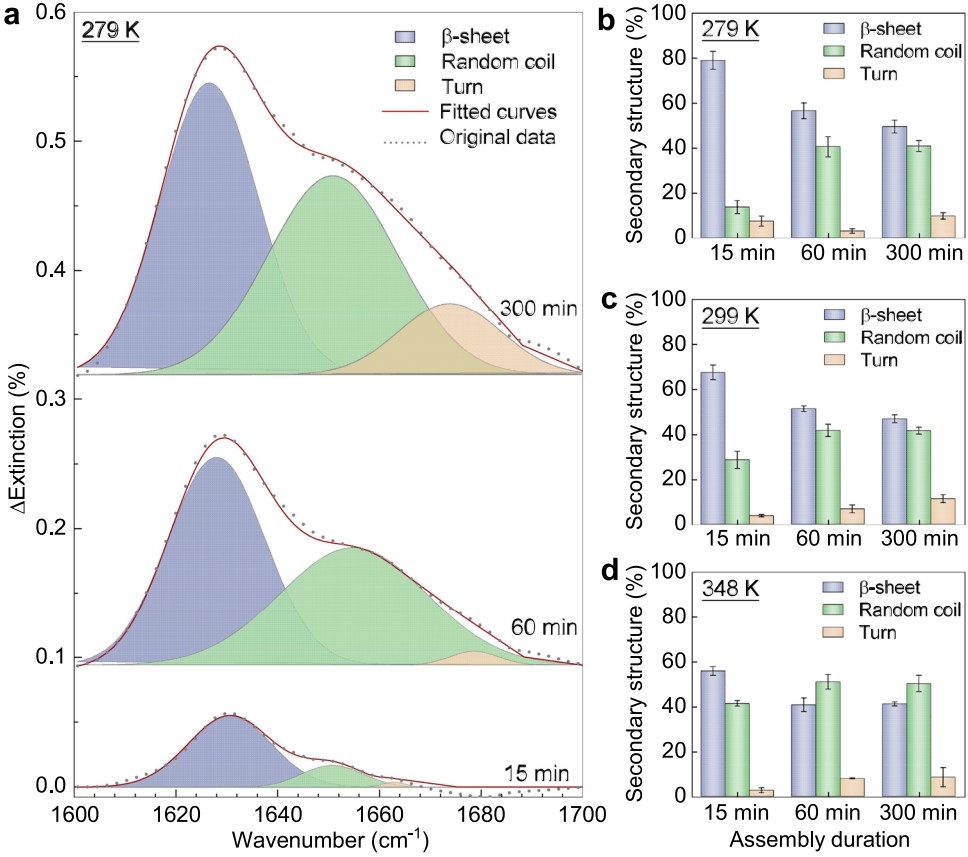

**Fig. 3 | Secondary structure changes during SNF assembly. a** ΔExtinction spectra changes of graphene plasmon-enhanced FTIR during SNF assembly at 279 K (details in Figs. S9 and S10). The extracted secondary structure content during the assembly at (**b**) 279 K, (**c**) 299 K (details in Figs. S11, S12), and (**d**) 348 K (details in Figs. S13, S14), respectively. The data of (**b**–**d**) were collected from four individual measurements of graphene plasmon-enhanced FTIR and expressed as mean values +/− SEM.

300 min, the SNFs became more enriched and elongated, resulting in longer and thicker SNFs with heights of ~4.8 nm and ~6.2 nm, respectively.

Figure 2d summarizes the heights of oligomers and SNFs assembled on graphene nanoribbons at different durations under varied temperatures (details in Fig. S6). As the temperature rises, the heights of the short rod-like SNFs (representing the combined heights of multiple stacked nanofibrils) also increase, and the lengths of the SNFs become longer, indicating a higher assembly rate. Remarkably, at lower temperatures, the elongation of SNFs proves notably challenging, likely due to the reduced thermal motion of protein molecules (details in Fig. S7). With the assistance of a deep learning algorithm (details in Note 2 of Supplementary Information), we extracted the lengths of assembled SNFs with a thickness of approximately 3–4 nm at different temperatures in Fig. S8. The average lengths of assembled SNFs is approximately 17 nm at 279 K, 24 nm at 299 K, and 95 nm at 348 K, respectively. This progressive increase in length with rising temperature further underscores the critical role of temperature in promoting SNF elongation.

**Secondary structure analysis during SNF assembly**
To correlate these assembly intermediates with different assembly stages, it is essential to assess their secondary structures. For example, based solely on their morphological features, clarifying whether the oligomer represents assembly intermediate in the nucleation stage or SF is challenging. This is because the observed oligomers exhibit a height similar to that of individual SF molecules. Therefore, we evaluate the β-sheet content of both oligomers and SNFs in relation to their morphology using graphene plasmon-enhanced FTIR. While

comprehensively describing SNF assembly presents a significant challenge, focusing on changes in β-sheet content offers a valuable perspective for understanding the fibrillation process. This approach is supported by extensive research that has demonstrably linked β-sheets to the remarkable mechanical properties of natural silks[2,45,46].

Figure 3a exemplifies the methods for extracting the secondary structure content of assembly intermediates at 279 K with different assembly durations. Here, ΔExtinction spectra are extracted by subtracting pristine Extinction spectra of SNFs with graphene plasmon enhancement from the baseline (details in Note 3 of Supplementary Information)[34]. The ΔExtinction spectra in the response range of the amide I band (1600–1700 cm⁻¹) are deconvoluted. The contributions to the amide I band mainly come from three distinct Gaussian peaks corresponding to β-sheet (i.e., 1616–1637 cm⁻¹), random coil (i.e., 1638–1662 cm⁻¹), and turn (i.e., 1663–1685 cm⁻¹)[47]. It is worth noting that the peak between 1685–1700 cm⁻¹, possibly belonging to intramolecular β-sheet, is too weak to be considered. Finally, the secondary structure content is calculated by determining the area percentage of each resolved peak.

As shown in Fig. 3b, we can extract the average content of secondary structures with standard error for semi-quantitative analysis: β-sheet is ~79% ± 4%, random coil is ~14% ± 3%, and turn is ~7% ± 2% at 15 min; β-sheet is ~57% ± 4%, random coil is ~40% ± 5%, and turn is ~3% ± 1% at 60 min; β-sheet is ~49% ± 3%, random coil is ~41% ± 2%, and turn is ~10% ± 1% at 300 min. The average secondary structure content, along with the standard error, are determined by applying the same fitting method to 3–4 extinction spectra (which were repeatedly measured for each sample, further details in Note 3)[48,49]. These results are depicted in Fig. 3b–d and Figs. S9–S14. To mitigate any potential errors

in secondary structure content that could arise from variations in fitting parameters (such as baselines, number of peaks, and the full width at half maximum of deconvolution peaks), we generated a range of fitting results. As shown in Fig. S15, there is a consistent decrease in β-sheet content with increasing assembly durations.

Therefore, the secondary structure content and its changes during the assembly process at 279 K, 299 K, and 348 K can be summarized in Fig. 3b–d, respectively. Remarkably, the oligomers with a height of 0.8 nm and 1.2 nm (i.e., after assembling for 15 min at 279 K and 299 K, corresponding to Fig. 2a, b) have a high β-sheet content of ~79% and ~67%. This is significantly higher than the β-sheet content of SF (details in Fig. S16). Since changes in secondary structure indicate assembly events[16,27,50], these observed oligomers are assembly intermediates rather than SFs. These experimental findings demonstrate that the oligomer has undergone a conformational transition from the amorphous SF to β-sheet-rich nuclei before SNF formation.

Correlating the morphology analysis in Fig. 2 with the corresponding changes in secondary structure from Fig. 3, we gain valuable insights into the SNF assembly process. Notably, the β-sheet-enriched oligomers formed on graphene at the nucleation stage (e.g., 15 min at 279 K and 299 K). The densely covered graphene by oligomers implies the high efficiency and fast pace of nucleation induced by graphene. As the assembly duration increases, the thickness of the assembly intermediates also increases, as demonstrated in Fig. 2d. This indicates that the assembly interface moves away from the graphene surface, leading to a reduced template effect and a consequent gradual decrease in the average β-sheet content. Furthermore, Figs. 3c and 2d (from 60 min to 300 min) illustrate that when the assembly interface is significantly distanced from graphene, the β-sheet content stabilizes, despite the increasing length and thickness of SNFs. It is also noteworthy that the β-sheet content of well-assembled SNFs at the graphene interface (300 min, 348 K) is comparable to those assembled in heated solutions, as detailed in Note 4 and Fig. S17 of the Supplementary Information. This suggests minimal influence of the graphene plasmonic infrared sensor on the growth stage of SNFs.

## Multi-scale MD simulations of SNF assembly

With a deep understanding of specific assembly intermediates, we have theoretically reproduced the assembly process through a combination of all-atom and coarse-grained MD simulations. Our primary focus is to revisit the dynamics of SNF assembly over extended timeframes while also examining the influence of environmental factors, specifically the graphene interface and temperature. Researchers have revealed that the C-terminus possesses a helical structure that typically stabilizes the protein against aggregation[2,22], whereas the N-terminus, being sensitive to pH changes, triggers the assembly process by forming β-sheet at acidic pH[39,51]. Since the degumming process disrupt the peptide chain of SF in experiment, the regenerated SF protein lacks C- or N- termini. Moreover, we mainly investigate the assembly of regenerated SF on graphene surfaces under neutral pH, and the C- and N-termini do not form β-sheets under neutral pH[2,51]. Thus, in our simulation models, a short repetitive peptide sequence as a fragment of SF without considering C- and N- termini (details in Methods and Note 5 of Supplementary Information for details) was adapted to investigate the formation of β-sheet. The initial structure of SF is generated using AlphaFold2[52].

To understand the initial stages of SF assembly on graphene (nucleation stage), we employed all-atom MD simulations. Since the real SF in solution exhibits a highly disordered conformation, we conducted 1 μs simulation to generate realistic SF models in solution. These models were then placed on the graphene surface, as shown in Fig. 4a. Subsequently, an additional 1 μs simulation was conducted to explore the assembly process of SF absorbed on graphene, as depicted in Fig. 4b. By capturing snapshots of SF at various time points (0 μs, 0.25 μs, 0.5 μs, and 1 μs), it becomes evident that SF undergoes

conformational transition, as indicated by the formation of β-sheet (highlighted by red circles) parallel to the graphene surface with the progression of assembly duration. Furthermore, a comparison of the assembly behavior of SF with and without graphene reveals that graphene can indeed accelerate the rate of SF conformational transitions. Additionally, at higher temperatures, thermal motion intensifies, leading to faster growth rates as well (details in Fig. S18). These insights align well with the experimental results.

As the assembly process of SNF extends into the growth stage, which is beyond the time and length scales accessible by all-atom MD simulations, we employed coarse-grained MD simulations. Two models were built to clarify the impacts by temperature and graphene-induced oligomers during the growth stage. Since first layer of oligomers covered the graphene surface, the first model starts with four graphene-template-induced oligomers (blue, β-sheet content ~51.2%), followed by introducing three new SF molecules (red), as shown in Fig. 4c. The second model begins with seven SF molecules to represent the assembly process in solution without graphene, as shown in Fig. S19a.

To investigate the temperature effect on facilitating SNF elongation, we compare the average contact number ($N$) based on first model at varied temperature (0.8\*$T$ and $T$, as described in Methods). As shown in Fig. 4c, the red molecules form well-ordered structure at 0.8\*$T$. Some red molecules tend to establish interfacial contacts along the geometric surface of their respective structures, resulting in a clearly defined contact interface (as magnified by a yellow dashed circle). It is noted that some red molecules exhibit no contact (as magnified by a black dashed circle), disrupting the assembly elongation process and resulting in shorter SNFs. Thus, red molecules have limited contacts ($N = 521 \pm 12$) with blue oligomers. In sharp contrast, when the temperature is increased to $T$, adjacent blue oligomers and red molecules undergo contact fusion promoted by random structures (as magnified by a green dashed circle). This results in an interconnected β-sheet network with numerous interfacial contacts ($N = 1175 \pm 15$), which provides the necessary stability for SNF elongation. Moreover, as shown in Fig. 4c and Fig. S19, the β-sheet peptide chain axis of the assembled oligomers tends to be parallel with the elongation axis of SNF at $T$, which is consistent with previous reports[53,54]. Therefore, the simulation results are consistent with experimental observations that lower temperatures are unbeneficial for SNF elongation.

MD simulations provide valuable insights into the structure-property relationships at the molecular scale. One key finding concerns the β-sheet content of the assembled structures. By summarizing β-sheet content in these two models, as shown in Fig. S19b, it is shown that the average β-sheet content experiences a decrease after the assembly process compared to the initial graphene-induced oligomers (51.2%). For instance, at 0.8\*$T$, the average β-sheet content drops to $49.2 \pm 0.3\%$. The reduction can be attributed to the newly added SF molecules (colored red) forming a β-sheet content of around 46.5% post-assembly, a structure closely matching that of the assembly formed at 0.8\*$T$ in model 2 without graphene (β-sheet = $46.4 \pm 0.3\%$). The observation underscores that the decrease in β-sheet content is a consequence of the diminishing template effect of graphene, aligning well with the experimental findings.

## Discussion

In summary, we introduce an innovative experimental approach utilizing graphene plasmonic infrared sensor, which can identify the secondary structure contents and corresponding morphologies of assembly intermediates (ranging from 0.8 nm to 6.2 nm in thickness) during SNF assembly. By combining multi-scale MD simulations, we gain insights into the molecular mechanisms underlying protein-protein and protein-environment interactions. The assembly process for regenerated *Bombyx mori* SF is thus provided as illustrated in

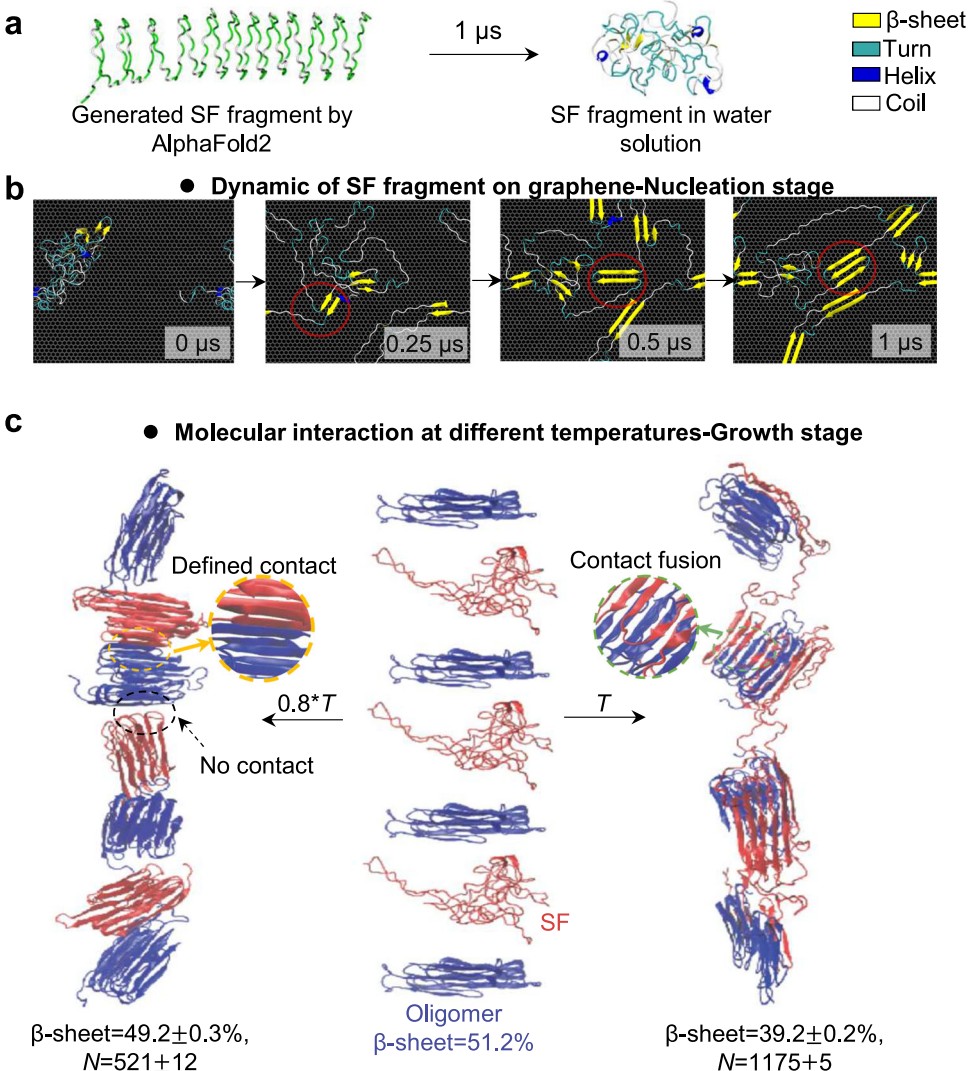

**Fig. 4 | Multi-scale MD simulations of SNF assembly. a** SF fragment model utilized in MD simulations. **b** Snapshots of SF fragment dynamics on graphene at nucleation stage (i.e., 0 μs, 0.25 μs, 0.5 μs, and 1 μs). The assembly snapshot of SF at different environment temperatures is shown in Fig. S17. The yellow, indigo, blue, and white parts represent β-sheet, turn, helix, and coil, respectively. **c** Snapshots of model 1 at the growth stage. Two environment temperatures (i.e., $0.8*T$ and $T$) are set. $N$ is the average contact number. The data were collected from five repeated simulations and expressed as mean values +/− SEM.

Fig. 1a. In nucleation stage, unfolded SF molecules undergo a conformational transition to form β-sheet-enriched oligomers. These oligomers then progress to the growth stage, connecting with adjacent molecules and elongating into short-rod-like SNFs. The β-sheet peptide chain axis of the assembled oligomers has a preferred orientation −parallel to the elongation axis of the SNFs. Ultimately, the short-rod-like SNFs align and elongate, becoming thicker and longer to form long-rod-like SNFs. Additionally, the critical roles of graphene-mediated interface and temperature in the SNF assembly process have been revealed. Specifically, the graphene interface accelerates SNF nucleation, while higher assembly temperatures enhance both SNF nucleation and elongation.

The discovery of β-sheet-enriched oligomers present prior to SNF formation aligns with the previous experimental observations of β-sheet-enriched nanocompartments in *Bombyx mori* silk gland[55], which motivates us to speculate about a potential pathway in the silk spinning process of *Bombyx mori*: During storage in the middle silk gland, SF has already assembled into β-sheet-rich oligomers and short-rod-like SNFs, which remain stable due to the presence of a hydrophilic coating. These β-sheet-dominated oligomers and short-rod-like SNFs can rapidly fuse into elongated nanofibrils during spinning. This is

essential for the rapid consolidation of silk under the constraints of limited spinning time and external stimuli.

Furthermore, we would like to highlight the advancements of the graphene plasmonic infrared sensor. By leveraging graphene plasmons, the sensor significantly enhances the sensitivity of the FTIR technique, enabling the detection of trace assembly intermediates with sub-nanometer resolution. In terms of practicality, the fabrication strategies of the graphene plasmonic infrared sensor are relatively standard (details in Methods)[56,57]. Additionally, the required equipment – commercially available micro-FTIR and AFM – is commonly found in most materials science and nanotechnology labs. Furthermore, the sensor's applicability extends to in-situ characterization of diverse SF assembly processes under complex environmental conditions. It can effectively analyze assembly dynamics in environments with varying pH levels, shear forces, ions, and solvents. This capability will prove beneficial for inspiring novel artificial spinning strategies.

## Methods
### The chemicals sampling
**Preparation of SF solution.** A standard protocol was followed to prepare the aqueous solution of pure *Bombyx mori* SF[41].

Firstly, removing sericin. A 2 L solution of $NaHCO_3$ with 0.5% *w/v* (weight/volume) was boiled; 10 g of silkworm cocoons were added and boiled for 30 min. The degummed silk was cleaned with distilled water and squeezed dry. Subsequently, it was boiled in a fresh $NaHCO_3$ solution for another 30 min to remove residual sercin. Then, it was dried overnight at room temperature or in an oven. Secondly, dissolving degummed silk fiber. The degummed silk fiber at a concentration of 10% *w/v* was dissolved in a 9.3 mol/L LiBr solution with a water bath at 333 K for 2 h. The solution was cooled to room temperature, and it was dialyzed in deionized water using a dialysis bag for 72 h to obtain a SF solution with a concentration of approximately 2% (*w/w*, weight in weight). After dialysis, the solution was centrifuged at 9600 × g (gravitational acceleration) at least twice for 30 min to remove insoluble impurities. The final SF solution was stored at 279 K.

### Fabrication of graphene plasmonic infrared sensor

First, the graphene (purchased from BEIJING GRAPHENE INSTITUTE CO., LTD) was transferred onto a 285 nm $SiO_2$/500 μm Si substrate (purchased from Silicon Valley Microelectronics, Inc) using the wet transfer method[44]. Then a 270 nm-thick electron beam resist (Polymethyl methacrylate, PMMA 950 K) film was spin-coated on graphene. Electron-beam lithography (Vistec 5000 + ES, 5-nm resolution, 100 keV beam) was used to pattern the graphene (~60 nm for the width and 2:1 for period-to-width) into arrays of nanoribbons. Subsequently, the patterned region was exposed after developing it with a PMMA developer (MiBK: IPA = 1:3). The exposed graphene was etched to create the desired nanoribbon structures using oxygen plasma (SENTECH). Electrode patterning was performed using electron-beam lithography (Vistec 5000 + ES) or photolithography (SUSS MA6). Electron beam evaporation (OHMIKER-50B) was utilized to deposit a 5 nm layer of chromium followed by a 50 nm layer of gold onto the graphene to create the electrodes. Finally, a lift-off process was conducted by immersing the sensor in acetone and alcohol to remove any remaining resist residues.

### Assembly process of SNF

To initiate the assembly process of SNF, both the graphene plasmonic infrared sensor and SF solution (concentration: 10.1 μg/mL) were preheated for 10 min at the desired temperature (e.g., 279 K, 299 K, 343 K). The sensor was then carefully immersed in the SF solution with the graphene surface facing downwards to prevent the deposition of SF molecules and impurities onto the graphene surface due to gravity. Following specific assembly durations, the sensor was thoroughly cleaned to remove loosely adsorbed assembly intermediates and prevent protein deposition. Cleaning involved rinsing the sensor with deionized water to eliminate any protein buildup caused by water evaporation from the SF solution. Finally, the sensor was dried by gently blowing nitrogen gas to avoid potential damage or contamination.

### Characterization of the graphene plasmonic infrared sensor

The morphologies of the assembled samples were characterized using AFM (Bruker Multimode8). FTIR transmission measurements were conducted using a Nicolet iN10 spectrometer equipped with an IR microscope and a 15X objective lens (Thermo Fisher). The aperture size for each measurement was set to 100 × 200 μm. The FTIR spectrometer was set with a resolution of 8 $cm^{-1}$, and an accumulation of 128 scans.

The electrical properties of the graphene plasmonic infrared sensor were characterized using a source meter (Keithley 2636B). The source meter allows for evaluating the electrical performance of every sensor. Additionally, the source meter was used to tune the back gate voltage, enabling control of electrical properties and modulation of the graphene plasmonic response.

### Molecular dynamics simulations

**All-atom molecular dynamics simulations.** AlphaFold2[52] was utilized to generate the structure of the SF fragment, serving as the initial configuration for MD simulations. Initial boxes with dimensions of approximately $10 × 10 × 10$ $nm^5$, each containing a SF fragment along with $3.2 × 10^4$ water molecules, were constructed using the PACKMOL program (Version 20.14.0)[58] and VMD 1.9.3 program[59]. The Amber ff14SB force field[60] was applied to SFs and graphene, and the OPC3 water model was used[61]. GROMACS 2021.7 code[62] was used to perform all MD simulations, while the VMD 1.9.3 program was used for visualizing trajectories. 1 μs simulations for both the simple solution box and the graphene interface model were performed, see Note 5 for simulation details. The last 0.2 μs of each trajectory was utilized for data extraction and analysis. Hydrogen bond criteria are defined as follows: distance (Donor-Acceptor) ≤ 0.35 nm ∨ angle (Hydrogen-Donor-Acceptor) ≤ 30°. The secondary structure was analyzed using the DSSP approach[63].

**Coarse-grained simulations.** A series of coarse-grained molecular dynamics simulations were conducted using OpenAWSEM to study the SF assembly using the AWSEM force field[64]. The same SF sequence as in all-atom MD simulations was employed. Each residue was simplified as three beads representing $C_\alpha$, $C_\beta$, and oxygen atoms, respectively. Simulations were initiated with a single SF fragment starting from a randomly extended chain, running for $2 × 10^6$ steps. The final structure of the oligomer was copied seven times for subsequent assembly simulations at two distinct temperatures: 0.8*T and *T*. *T* = 360 K. However, due to the usage of the coarse-grained model, this value does not correspond to 360 K in reality. So we used 0.8*T and *T* when referring to the coarse-grained simulation results. For each, the last $10^7$ steps were run and repeated 5 times with final snapshots as shown in Fig. 4c. The β-sheet content of the final state was calculated using DSSP[65]. Contact is defined as intermolecular residue pairs between two adjacent SFs when the distance between $C_\alpha$ atoms is less than 10 Å.

### Reporting summary

Further information on research design is available in the Nature Portfolio Reporting Summary linked to this article.

## Data availability

Source data are provided with this paper.

## Code availability

The code that support the findings of this study are available in the Source Data file.

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

## Acknowledgements

This work was supported by the National Key R&D Program of China (2021YFA1201500, Q.D.; 2022YFC3400501, Q.W.), the National Natural Science Foundation of China (51925203, Q.D.; 52022025, X.Y.; 21935002 and 52322305, S.L.; 52102160, X.G.; 51972074, X.Y.; 32000882, Q.W.; and U2032206, Q.D.), the Strategic Priority Research Program of the Chinese Academy of Sciences (XDB36000000, Q.D.), the Chinese Academy of Sciences Project for Young Scientists in Basic Research (YSBR-086, X.Y.), Youth Innovation Promotion Association C.A.S. (X.Y.), Beijing Natural Science Foundation (2234095, C. W.) and the Special Research Assistant Program of the Chinese Academy of Sciences (X.G.).

## Author contributions

The concept for the experiment was initially developed by Q.D., X.Y., S.L., and Q.W.; Graphene plasmonic infrared sensors were designed and prepared by C.W. assisted by Y.D., S.Z., and X.D.; The FTIR experiments were performed by Y.D. and C.W. under the direction of Q.D., and X.Y.; Molecular simulations were performed by Q.W., Y.H., S.L., and L.Y.; Silk fibroin solution was prepared and characterized by Y.D., and C.Z.; Experimental data processing and analysis was performed by C.W. and Y.D. assisted by X.G., C.J., and P.M.; C.W., Q.D., X.Y., and Y.D. co-wrote the manuscript with inputs and advice from S.L. and Q.W.; C.W. and Y.D. contributed equally to this work. All authors discussed the results at all stages and participated in the development of the manuscript.

## Competing interests

The authors declare no competing interests.
