## [Peer Review File · Nature Communications]

In-situ Observation of Silk Nanofibril Assembly via Graphene plasmonic infrared sensorREVIEWER COMMENTS

Reviewer #1 (Remarks to the Author):

I appreciate the work of Wu et al., with respect to methodological advances. Indeed, the introduction of graphene plasmons enhanced FTIR is a significant development in sensitivity for silk and other biomolecules conformational transformations. The quality of AFM imaging and MD modelling is excellent. I have, however, some reserves with respect to the samples and the simulations.

Specific remarks:

- The nature and homogeneity of silk fibroin (SF) prepared by dissolution of cocoons and dialysis is not established. I am not convinced that the sample consisted only of the fibroin molecules shown schematically in Figure 1. The authors should indicate how they have characterized the fibroin solution or provide corresponding literature references. This is important as the kinetics and pathway of assembly could be influenced by molecular fragments.
- The graphene surface cannot be compared with the complexity of a bio-spinning duct, involving changes in pH, ionic flow and a shearing gradient (Vollrath et al., *Nature* (2001) 410, 541...). Indeed, β -sheet transformation is assumed occurring under action of the shearing gradient. Performing MD simulations based on uncharacterized, single fibroin molecules and oligomers cannot be consoled with processes in the spinning duct. I do not exclude that assembly-processes in certain artificial spinning set-ups -using aqueous environments- could be modeled in this way. It appears to me, however, that the most advanced (in terms of mechanical properties achieved) artificial silk spinning set-ups try to mimic the environment of a bio-spinning duct. Probing amyloid nanofibrillation might be a much better target for the methods developed by Wu et al., Another possibility would be probing nanofibrillation in an evaporating droplet.
- The involvement of the C- and N-terminus in the assembly process is not considered by the authors but has been shown to be relevant for silk assembly (Askarieh et al., *Nature* (2010) 465, 236...)
- According to a model of fibroin self-assembly (Yoshioka et al., *Nat. Com.* (2019) 10, 1469...) the fibroin molecular chain can be thought as a repeating motif of poly(ala) and non-poly(ala) blocks. In the silk gland, poly(ala) is thought to have a helical conformation. Neighboring helical poly(ala)s are assumed aggregating into a bundle. In the spinning ducts shearing gradient, poly(ala) and part of the non-poly(ala) blocks transform into β -sheet conformation, resulting in nanofibrils. This model differs from the model used for the MD simulations (Figure 4).
- The lateral dimensions of crystalline β -sheets in fibroin nanofibrils are known from X-ray diffraction. How do these compare to contact fusion self-assembly?

Reviewer #2 (Remarks to the Author):

This paper studies the transition from silk fibroin (SF) to silk nanofibrils (SNF). To do so, the authors use a combination of graphene plasmonic infrared sensor measurements and molecular dynamics (MD) simulations. The main conclusion of this study is that SFs changes from an amorphous conformation to β -sheet-rich oligomers, and this transition can be modulated by temperature. β -sheet-rich oligomers subsequently interact with other amorphous SF, eventually forming SNFs. This process is associated with a gradual reduction in β -sheet content.

I have concerns about the suitability of a graphene-based technique to help in elucidating how the SNF assembly process happens in Nature. Earlier studies have (see e.g., <https://doi.org/10.1021/acsami.5b05615>) have shown that graphene can severely disrupt β -sheets in SF, due to a competition for hydrophobic interactions. What is the authors view on this point? Could this influence the significance of the data reported in this manuscript?

The manuscript also incurs in some scientific inaccuracies. The authors claim that "SFs consist of highly repeated alternating hydrophobic sequences (i.e., GAGAGS) and hydrophilic sequences (i.e., GAGAGY), which tend to assemble by forming β -sheets and random coils, respectively"). However, the references used to support that statement (48 and 49) do not indicate that for GAGAGY. In fact, both sequences are hydrophobic and prone to form β -sheets (see e.g., <https://doi.org/10.1021/bm501667j> or <https://doi.org/10.1016/j.actbio.2014.09.013>, or even reference 49 in this work). Furthermore, Figure 1 also indicates that GAGAGY forms random coil regions in SF, which is incorrect.

Thus, I cannot recommend this manuscript for publication in Nature Communications.

Here also some specific comments about the text:

- The use of deconvoluted FTIR spectra to quantify secondary structures in protein materials is controversial. Numerous methods have been proposed, and the results from those methods vary widely depending on the baseline selection, number of peaks used for deconvolution, etc. Therefore, many authors prefer to use this method as a qualitative analysis, rather than as an absolute quantification of the secondary structure. To that end, would the authors be able to provide some statistics for the percentages reported on lines 182-184? This would help in understanding the accuracy/variability of the values reported in this manuscript.
- Lines 187-190: "the oligomers with a height of 0.8 nm and 1.2 nm [...] have a high β -sheet content of $\sim 79\%$ and $\sim 67\%$, respectively. This indicates that the oligomers are the assembly intermediates rather than SFs": What is the basis for this statement? Can the authors explain in more detail why a higher β -sheet content is indicative of those structures being intermediates? How much higher are those β -sheet contents, compared to amorphous SF?
- Line 202: β -sheets in silk materials are stable and durable structures (materials with high β -sheet contents can last for extremely long periods). However, the authors report a decrease in β -sheets during the SNF formation. Do the authors have any explanation for how the β -sheets present in oligomers might be disrupted, despite the high stability of such structures?
- Furthermore, the authors use MD simulations to claim that SF in contact with graphene forms more β -sheets than in water. However, I doubt that the MD simulations reported here are likely to show the formation of β -sheets in water-only systems. The timescale for the formation of β -sheets is unlikely to be captured by MD simulations, especially if no replica-exchange sampling is used. I think that the simulations reported here can only be used to demonstrate that SF has a high tendency to adsorb on graphene.
- Discussion: The authors indicate that SF is already stored in the glands of silkworms as a solution of oligomers rich in β -sheets that is subsequently spun into fibers. However, the current understanding of this process is that spinning induces the formation of even more β -sheets due to the alignment of fibers. In contrast, this paper shows a decrease in β -sheets. I am therefore not sure how the results here reported could help in elucidating the formation of SNFs in Nature, which is the main aim of this paper as indicated by the authors. Could the reduction in β -sheets be caused by the hydrophobicity of graphene?
- Lines 301-302: "To obtain a 9.3 mol/L SF solution, degummed silk at a concentration of 10% w/v was dissolved in a LiBr solution". Can the authors confirm that this is the method that they used? Based on my experience on common silk degumming methods, the 9.3 M solution would be from LiBr, not from SF. I believe there is an error in this sentence.
- Line 329-331: "the sensor was cleaned thoroughly by rinsing it with deionized water to remove any loosely adsorbed assembly intermediates. The sensor was dried by gently blowing nitrogen gas to ensure no residual moisture remained". Since SF can hydrophobically adsorb on graphene, would it be possible for the authors to provide evidence that this cleaning procedure is sufficient to remove adsorbed SF from the graphene surface?
- Note 4, SI: at what distance do the authors place the SF polypeptide on top of the graphene

surface for MD simulations?

Some minor typos:

- Line 108: "excite", not "excited".
- Line 131: "involves", not "involve".
- Line 246: "it exhibits" instead of "there exhibits".

Reviewer #3 (Remarks to the Author):

This manuscript reports in situ measurement of silk nanofibril assembly using graphene nanoribbon-based FTIR sensing. The authors prepared silk fibroin from silkworm cocoons and assembled silk fibroin to silk nanofibrils on a patterned graphene substrate, which is also used as a FTIR sensing substrate, under different temperature and duration to understand silk nanofibril assembly mechanisms. Surface morphologies and FTIR spectra in the process of nanofibril assembly are mainly discussed, and molecular dynamics simulations were used to support experimental results. The reviewer has concerns regarding materials characterizations and design of FTIR (see comments 2, 3) and the clarity in the discussions about the novelty of the work (see comments 1, 4).

1. The reviewer thinks the title does not represent the work reported here because it is not specific enough. The authors may want to consider changing the title.

2. The authors used patterned graphene as a substrate to enhance the silk nanofibrils assembly as well as for graphene plasmon-enhanced FTIR.

2-1. Graphene plasmon-enhanced FTIR: plasmonic properties of graphene are highly depending on the geometry and length scale of graphene patterns. The authors mentioned that "the graphene nanoribbon widths were designed to excited graphene plasmons in proper frequency range," in the Results section, but any further details are missing. Based on the AFM images in Figure 2, the width of the nanoribbon can be roughly guessed, but the reviewer believes that more details should be provided. Specifically, why did the authors decide to have roughly 75 nm for the width and 75 nm for the distance between ribbons? What are corresponding plasmonic properties/effects from such patterns? Please add more details with relevant references and/or simulation results to support the structure used in this work.

2-2. Material: The authors specified that they purchased graphene from a vendor. Was the graphene monolayer or thicker? Either way, please provide materials characterization results as the number of layers of 2D materials has a significant impact on the properties, the reviewer believes that material characterization results should be presented even though it was a commercially available product.

2-3 Electrical characterization of graphene plasmonic infrared sensor: In the subsection of "Characterization of the graphene plasmonic infrared sensor" of the Methods section, the authors discussed: "The electrical properties of the graphene plasmonic infrared sensor were characterized using a source meter (Keithley 2636B). The source meter allows for the evaluation of electrical performance of every sensor. Additionally, the source meter was used to tune the back gate voltage, enabling control of electrical properties and modulation of the graphene plasmonic response." However, there is no data or discussion related to these electrical characterizations in both manuscript and supporting information. Additionally, if the authors tuned graphene's plasmonic response by applying electric fields, the specific conditions should be discussed somewhere in the manuscript as well.

3. Figure 2d: Figure 2d presents the height change of the silk nanofibril structures formed on graphene nanoribbons. The reviewer was not totally convinced that long-rod-like silk nanofibrils would have a height higher than oligomers and short-rod-like silk nanofibrils. Longer fibrils may be stacked more efficiently and thus may have a lower height, compared to oligomers and short-rod-like fibrils, as described in the small schematic illustrations inserted in the panel d. The reviewer suggests that the authors discuss a bit more details about the height comparisons for future

readers.

4. The reviewer was curious about the new insight the authors revealed from this work. Specifically, was this the first paper reporting in situ measurement of silk nanofibril assembly? Was this the first paper using a graphene plasmon-enhanced FTIR sensor to study silk nanofibril assembly? Was this the first paper having silk nanofibril assembly on a graphene substrate? The authors mentioned in the abstract that "this work's results offer a detailed understanding of the silk nanofibril's assembly mechanism," but it is not clear which part of the mechanism is newly reported. Please discuss the novelty of your work clearly to help your readers understand your work better.

5. Nanofibril assembly temperature: The nanofibril assembly process was measured in different temperatures including 279 K, 299 K, and 348 K. 348 K showed the fastest assembly. As 348 K seems very high for a living organism, the reviewer was wondering at which temperature a silkworm produces its silk.

6. The authors use the term "assembly velocity" multiple times in the manuscript. However, it sounds strange to me, as velocity is a vector property. Could you use "assembly rate" or another term?

Reviewer #4 (Remarks to the Author):

The manuscript by Wu et al. studies the assembly process of silk nanofibrils (SNFs) using graphene ribbon plasmonic infrared sensor. They monitor conformational transitions of silk fibroin to SNF and proposes an assembly model. Then the authors discuss the potential implications for artificial silk spinning techniques.

While the study presents some interesting understanding of the assembly mechanism of SNFs, this reviewer finds that novelty is lacking to publish this in Nature Communications. The research uses graphene ribbon system, which is by now well-established for sensing applications. Specifically, their focus on the graphene nanoribbon sensor's application in silk fibril assembly in a dry environment is too narrow in scope, not generally applicable to a wide range of proteins, and is not novel any more.

Finally, the implications of the study, primarily centered on silk spinning techniques, may not resonate strongly with the broad readership of Nature Communications.

I suggest that the authors submit this manuscript to a more specialized journal.

RESPONSE TO REVIEWERS' COMMENTS

We appreciate the insightful and constructive comments raised by the reviewers. Here, we have addressed all the comments. Our responses are shown in blue, and our revisions to the manuscript and supplementary materials are indicated in red.

Reviewer #1 (Remarks to the Author):

Comment: I appreciate the work of Wu et al., with respect to methodological advances. Indeed, the introduction of graphene plasmons enhanced FTIR is a significant development in sensitivity for silk and other biomolecules conformational transformations. The quality of AFM imaging and MD modelling is excellent. I have, however, some reserves with respect to the samples and the simulations.

Reply: We appreciate the positive evaluation of the novelty of our work and the valuable suggestions for further improvement.

Specific remarks:

1. The nature and homogeneity of silk fibroin (SF) prepared by dissolution of cocoons and dialysis is not established. I am not convinced that the sample consisted only of the fibroin molecules shown schematically in Figure 1. The authors should indicate how they have characterized the fibroin solution or provide corresponding literature references. This is important as the kinetics and pathway of assembly could be influenced by molecular fragments.

Reply: We are grateful to the reviewer for highlighting this important aspect. In our research, we adhered to a **well-established protocol for preparing aqueous solutions of pure *Bombyx mori* silk fibroin** (Nature, 2003, 424, 1057–1061; Nature Protocols, 2011, 6, 1612–1631). The protocol is stated in Methods of revised manuscript (Line 336-348 on Page 12) “*Firstly, removing sericin: A 2 L solution of NaHCO₃ with 0.5% w/v (weight/volume) was boiled; 10 g of silkworm cocoons were added and boiled for 30 minutes... After dialysis, the solution was centrifuged at 9600×g (gravitational acceleration) at least twice for 30 minutes to remove insoluble impurities.*”, which can get pure fibroin solution. This method has been extensively employed to get regenerated silk fibroin (e.g., Nature Communications, 2022, 13, 7856; Nature Nanotechnology, 2021, 16, 1342–1348; Acta Biomaterialia, 2013, 9(8): 7806-7813) and has demonstrated the purity of the obtained silk fibroin (Macromol. Biosci. 2008, 8: 1006-1018). We have added related references.

Changes made in the revised manuscript:

We thank the reviewer for this question, in response to which we have made the following changes in Lines 92-95 on Page 4 of the revised manuscript:

“**The assembly of regenerated SF in solutions serves as a widely used in vitro model for exploring the mechanism of silk spinning^{48,49}.** Thus, we utilized **regenerated SFs extracted** from

Bombyx mori silkworm cocoon⁵⁰ to assemble on the graphene plasmonic infrared sensor (see Methods).”

2. The graphene surface cannot be compared with the complexity of a bio-spinning duct, involving changes in pH, ionic flow and a shearing gradient (Vollrath et al., Nature (2001) 410, 541...). Indeed, β -sheet transformation is assumed occurring under action of the shearing gradient. Performing MD simulations based on uncharacterized, single fibroin molecules and oligomers cannot be consoled with processes in the spinning duct. I do not exclude that assembly-processes in certain artificial spinning set-ups -using aqueous environments- could be modeled in this way. It appears to me, however, that the most advanced (in terms of mechanical properties achieved) artificial silk spinning set-ups try to mimic the environment of a bio-spinning duct. Probing amyloid nanofibrillation might be a much better target for the methods developed by Wu et al., Another possibility would be probing nanofibrillation in an evaporating droplet.

Reply: We are grateful for the reviewer's suggestion. We concur that tracking the assembly of silk fibroin into silk fibers in natural environment is the most essential condition for understanding the natural spinning mechanism. *Bombyx mori* silk production involves the transformation of silk fibroin from a soluble state to insoluble silk fibers. The initial stage encompasses the synthesis (in posterior silk gland) and storage of silk fibroin (in middle silk gland) in a soluble state at a very high concentration. The subsequent stage, known as spinning, is triggered during the passage of silk fibroin through the anterior silk gland, where a combination of physical and chemical stimuli, including variations in pH, ionic flow, and shear gradients, induces fibroin assembly into silk fibers. Although the most advanced artificial spinning technologies try to mimic the environment of anterior silk gland in silkworm (or duct in spider), the resultant fiber properties still fall short of those exhibited by natural silk.

This prompts us to focus on the assembly behaviors of silk fibroin during its storage, which is crucial to ensure the rapid and successful continuation of the spinning process (Nature, 2003, 424, 1057–1061; Nature 2010, 465, 239–242; Int. J. Mol. Sci. 2016, 17(8), 1290; Nature Communications, 2022, 13, 7856). However, due to the lack of in-situ characterization methods, the intricate mechanism that governs the silk nanofibril assembly, which involves both conformational transitions and the conjunction of protein molecules has never been studied.

In our study, we develop a graphene plasmonic infrared sensor that can link morphology and secondary structure of nanoscale intermediates during the silk fibroin assembly. The insights into the dynamics of regenerated silk fibroin over extended molecular timeframes bridges a significant knowledge gap in the transformation from silk fibroin to silk nanofibril. Particularly, the discovery of β -sheet-enriched oligomers present prior to the silk nanofibril formation motivates us to speculate about a potential storage pathway in the natural silk fibroin storage.

We gratefully acknowledge the reviewer's insightful recommendation to broaden our research to encompass various protein assembly systems via our methodology. This extension includes the exploration of amyloid nanofibrillation and the nanofibrillation processes within evaporating droplets. Considering these systems undergo assembly processes akin to those observed in the middle silk gland, our methodology proves equally applicable. In the future, we will endeavor to investigate these protein assembly processes further.

Changes made in the revised manuscript:

We thank the reviewer for this question. We have rewritten the speculation regarding the natural assembly mechanism from the Conclusion section of the revised manuscript (Lines 307-316):

“Particularly, the discovery of β -sheet-enriched oligomers present prior to SNF formation aligns with the previous experimental observations of β -sheet-enriched nanocompartments in silk gland⁶², which motivates us to speculate about a potential pathway in the silk spinning process: During storage in the middle silk gland, SF has already assembled into β -sheet-rich oligomers and short-rod-like SNFs, which remain stable due to the presence of a hydrophilic coating and regulation by ions, pH, and N- and C-termini nanocrystals. These β -sheet-dominated oligomers and short-rod-like SNFs can rapidly fuse into elongated nanofibrils when they are extruded during silk spinning, and the formation of more β -sheets might be induced due to the alignment of fibers. This is essential for the rapid consolidation of silk under the constraints of limited spinning time and external stimuli.”

3. The involvement of the C- and N-terminus in the assembly process is not considered by the authors but has been shown to be relevant for silk assembly (Askarieh et al., Nature (2010) 465, 236...)

Reply: We are thankful to the reviewer for highlighting this crucial issue. We concur that C- and N-terminus are relevant for silk assembly. The C-terminus typically stabilizes the protein against unwanted aggregation and has a helix structure (Nature, 2010, 465, 239–242; Nature Review Materials, 2018, 3, 18008; International Journal of Biological Macromolecules, 2021, 169: 473-479). The N-terminus, sensitive to pH changes, triggers the assembly process when the pH decreases. At a neutral pH, the N-terminus prevents premature β -sheet formation by maintaining a random coil conformation (Nature, 2010, 465, 236-238; Journal of Molecular Biology, 2012, 418(3): 197-207; Biomacromolecules 2017, 18, 8, 2521–2528). However, the C- and N-terminus are not considered due to the following two reasons:

(1) Our experiments, conducted at neutral pH, suggest that the C- and N-termini do not contribute to form β -sheets. Therefore, our study primarily concentrated on the assembly process of repetitive sequences that can form β -sheet on graphene.

(2) We utilized regenerated silk fibroins to study silk nanofibril assembly on graphene. The molecular weight of this regenerated silk fibroin is around 160 kDa (detailed in Figure S1), which is much less than the molecular weight (>375 kDa) found in natural silk fibroin (Nature,

2003, 424, 1057–1061). As a result, most molecules in the solution lack N- or C-termini.

Nevertheless, while C- and N- terminus are essential for regulating the assembly process of silk nanofibril, we also recognize the need for further research on the N- and C-termini's regulatory roles of silk assembly by introducing purified termini constructs and specific environment changes.

Changes made in the revised manuscript:

We thank the reviewer for this question, in response to which we have made the following changes in Lines 232-238 on Page 8-9 of the revised manuscript:

“Researchers have revealed that the C-terminus possesses a helical structure that typically stabilizes the protein against aggregation^{9, 28, 59}, whereas the N-terminus, being sensitive to pH changes, triggers the assembly process by forming β -sheet at acidic pH^{48, 60, 61}. Since we mainly focus on the assembly of reconstituted SF on graphene surfaces under neutral pH conditions, the C- and N-termini do not form β -sheets^{9, 28, 58}. In our simulation models, a short peptide sequence without considering C- and N- termini as a fragment of SF (details in Methods and Note 4 of Supplementary Information for details) is adapted.”

The molecular weight data of silk fibroin is supplemented in Figure S1 in revised supplementary material:

Supplementary Figure S1 Characterization the molecular weight (MW) of silk fibroin (SF). (a) Polyacrylamide gel images of SF with reference protein ladder. The SF solution, with

a concentration of 1.65 wt%, was blended with loading buffer and subjected to electrophoresis at 120 V for 80 minutes. (b) The relation between the log (MW) and the distance determined through simple linear regression analysis ($R^2=0.9929$). This analysis yielded a linear equation, which was then applied to convert the distances measured in inches on the gel images to molecular weight (MW) values for the sample lanes. (c) The MW distributions of SF with highest intensity peak at 152 kDa and mean MW of 160 kDa. This distribution is broad due to the SF molecules in the spinning stock solution forming a fractal network structure.

4. According to a model of fibroin self-assembly (Yoshioka et al., Nat. Com. (2019) 10, 1469...) the fibroin molecular chain can be thought as a repeating motif of poly(ala) and non-poly(ala) blocks. In the silk gland, poly(ala) is thought to have a helical conformation. Neighboring helical poly(ala)s are assumed aggregating into a bundle. In the spinning ducts shearing gradient, poly(ala) and part of the non-poly(ala) blocks transform into β -sheet conformation, resulting in nanofibrils. This model differs from the model used for the MD simulations (Figure 4).

Reply: We are grateful to the reviewer for raising this insightful question. Silk produced by different worm species varies in amino acid sequence. Our study focuses on silk from the *Bombyx mori* silkworm cocoon, which has a distinct composition. It doesn't have repeat motif of poly(ala), but predominantly consists of repeating motifs such as GAGAGS and GAGAGY, as established in previous studies (Biomacromolecules, 2005, 6, 5, 2563–2569). The research by Yoshioka et al. (Nature Communications, 2019, 10, 1469) examines *E. variegata* bagworm silk, which has a repeating motif of poly(ala) and non-poly(ala) blocks. This difference in amino acid sequence is crucial in our experimental and simulation studies, as it significantly impacts the silk's properties and behavior. Moreover, recent research indicates potential conformational transitions in the silk gland, prior to reaching the bio-spinning duct (Nature Communications, 2022, 13, 7856). This also highlights the importance of in-situ investigation into the assembly process of silk fibroin to silk nanofibril in vitro, encompassing both conformational transitions and protein molecule conjunctions.

5. The lateral dimensions of crystalline β -sheets in fibroin nanofibrils are known from X-ray diffraction. How do these compare to contact fusion self-assembly?

Reply: We are grateful for the reviewer's question. Indeed, XRD can provide such dimensional information. Literatures show that the dimensions of β -sheets in natural silk are about $0.938 \times 0.949 \times 0.698 \text{ nm}^3$ (e.g., International Journal of Biological Macromolecules, 1999, 24, 127). In contrast, silk nanofibrils formed through thermally induced self-assembly are reported to have dimensions of approximately $1.06 \times (0.92-1.05) \times 0.76 \text{ nm}^3$ (Macromolecular Rapid Communications, 2021, 42(3), 12000435). However, our study faces a challenge in analyzing silk nanofibrils assembled on the graphene surface by using XRD. The main issue is the small size of these nanofibrils, with a maximum height of less than 10 nm,

making such crystallographic analysis particularly demanding.

Reviewer #2 (Remarks to the Author):

This paper studies the transition from silk fibroin (SF) to silk nanofibrils (SNF). To do so, the authors use a combination of graphene plasmonic infrared sensor measurements and molecular dynamics (MD) simulations. The main conclusion of this study is that SFs changes from an amorphous conformation to β -sheet-rich oligomers, and this transition can be modulated by temperature. β -sheet-rich oligomers subsequently interact with other amorphous SF, eventually forming SNFs. This process is associated with a gradual reduction in β -sheet content.

(1) I have concerns about the suitability of a graphene-based technique to help in elucidating how the SNF assembly process happens in Nature. Earlier studies have (see e.g., <https://doi.org/10.1021/acsami.5b05615>) have shown that graphene can severely disrupt β -sheets in SF, due to a competition for hydrophobic interactions. What is the authors view on this point? Could this influence the significance of the data reported in this manuscript?

Reply: We express our gratitude to the reviewer for highlighting these important questions. In our experiments, the oligomers with heights of 0.8 nm and 1.2 nm exhibit a higher β -sheet content (>50%) compared to that of silk fibroin and well-assembled silk nanofibrils without graphene (details provided in Fig. 3 and Figure S15). The densely covered graphene by oligomers implies the high efficiency and fast pace of nucleation induced by the template effect of graphene. As the assembly duration increases, the thickness of the assembly intermediates also increases, as demonstrated in Fig. 2d. This indicates that the assembly interface moves away from the graphene surface, leading to a reduced template effect and a consequent gradual decrease in the average β -sheet content, which is consistent with our experimental results in Figure 3. Furthermore, Fig. 3c and Fig. 2d (from 60 min to 300 min) illustrate that when the assembly interface is significantly distanced from graphene, the β -sheet content stabilizes, despite the increasing length and thickness of SNFs. The β -sheet content of well-assembled SNFs at the graphene interface (300 min, 348 K) is comparable to those well-assembled in heated solutions, as detailed in Note 3 and Fig. S16 of the Supplementary Information. Consequently, graphene facilitates the nuclear of β -sheet, and does not disrupt the β -sheet content of SNFs that now directly contact with it.

In the literature mentioned by the reviewer (ACS Applied Materials & Interfaces 2015, 7, 39, 21787–21796), Cheng et al utilized a MD simulation to show that graphene can disrupt β -sheets in silk fibroin in 50 ns. It is noted that the conformational transition will continue after 50 ns. In the longer timescale simulations within 1 μ s, we observed that graphene initially induces the hydrophobic core of silk fibroin to unfold on its surface, which might disrupt the β -sheets in silk fibroin. Subsequently, β -sheets rapidly form due to graphene's template effect. The template effect is due to the combination effect of hydrophobic interaction, π - π stacking,

electrostatic interaction, and etc. There is a complex interaction between graphene and silk fibroin across long timescale. The experimental evidence demonstrates that the graphene interface does not inhibit the formation of β -sheets; on the contrary, it facilitates their growth.

Changes made in the revised manuscript:

We thank the reviewer for this question, in response to which we have made the following changes in Lines 215-226 on Page 8 of the revised manuscript:

“As the assembly duration increases, the thickness of the assembly intermediates also increases, as demonstrated in Fig. 2d. This indicates that the assembly interface moves away from the graphene surface, leading to a reduced template effect and a consequent gradual decrease in the average β -sheet content. Furthermore, Fig. 3c and Fig. 2d (from 60 min to 300 min) illustrate that when the assembly interface is significantly distanced from graphene, the β -sheet content stabilizes, despite the increasing length and thickness of SNFs. It is also noteworthy that the β -sheet content of well-assembled SNFs at the graphene interface (300 min, 348 K) is comparable to those assembled in heated solutions, as detailed in Note 3 and Fig. S16 of the Supplementary Information. Consequently, graphene plasmonic infrared sensor has minimal influence on the growth stage of SNFs.”

(2) The manuscript also incurs in some scientific inaccuracies. The authors claim that “SFs consist of highly repeated alternating hydrophobic sequences (i.e., GAGAGS) and hydrophilic sequences (i.e., GAGAGY), which tend to assemble by forming β -sheets and random coils, respectively”). However, the references used to support that statement (48 and 49) do not indicate that for GAGAGY. In fact, both sequences are hydrophobic and prone to form β -sheets (see e.g., <https://doi.org/10.1021/bm501667j> or <https://doi.org/10.1016/j.actbio.2014.09.013>, or even reference 49 in this work). Furthermore, Figure 1 also indicates that GAGAGY forms random coil regions in SF, which is incorrect.

Thus, I cannot recommend this manuscript for publication in Nature Communications.

Reply: We appreciate the reviewer for raising these concerns. We apologize for missing the citation of references (e.g., Biophysical Journal, 2000, 78(5): 2690-2701; Chemical Communications, 2009, 7506-7508; ACS Materials Letters, 2020, 2, 2, 153-160) that raise the opinion: GAGAGY is hydrophilic and prone to form random coils. We also acknowledge the literature mentioned by the reviewer (e.g., Acta Biomaterialia, 2015, 11: 212–221; Biomacromolecules, 2015, 16(2): 606–614), which presents a contrary opinion. This has prompted us to conduct a thorough literature review. It was noted that these differing opinions are based on certain references (e.g., Mater. Sci. Eng. C, 2001, 14: 41–46; Proteins: Structure, Function, and Genetics, 2001, 44(2): 119-122; J. Mol. Evol., 1994, 38: 583–592; Trends Biochem. Sci., 1982, 7: 105–108). Unfortunately, we did not find experimental or theoretical data to directly prove GAGAGY repeats are hydrophilic or hydrophobic and form β -sheet in these references or other literature. However, it is shown that GAGAGY repeats tend to form random coils rather than β -sheets, according to direct experimental evidence presented in the

literature (Soft Matter, 2013, 9: 11325-11333). Given the debate over the properties of GAGAGY, we have removed the related description to avoid unnecessary discussion. We believe this revision will not affect the significance or novelty of our manuscript.

Changes made in the revised manuscript:

We thank the reviewer for this question, in response to which we have made the following changes in Lines 92-94 on Page 4 of the revised manuscript:

“The assembly of regenerated SF in solutions serves as a widely used in vitro model for exploring the mechanism of silk spinning^{48,51}. Thus, we utilized regenerated SFs extracted from Bombyx mori silkworm cocoon to assemble on the graphene plasmonic infrared sensor (see Methods).”

In addition, we change Figure 1a in the revised manuscript as well:

Here also some specific comments about the text:

(3) The use of deconvoluted FTIR spectra to quantify secondary structures in protein materials is controversial. Numerous methods have been proposed, and the results from those methods vary widely depending on the baseline selection, number of peaks used for deconvolution, etc. Therefore, many authors prefer to use this method as a qualitative analysis, rather than as an absolute quantification of the secondary structure. To that end, would the authors be able to provide some statistics for the percentages reported on lines 182-184? This would help in understanding the accuracy/variability of the values reported in this manuscript.

Reply: We express our gratitude for the reviewer's suggestion. We agree that the specification of secondary structure content in proteins through infrared spectroscopy fitting can be influenced by the selected fitting parameters. To avoid any inaccuracy introduced by the quantitative values, we only semi-quantitative analyze the trend by statistically collecting secondary structure content in the assembly duration. It is noticed that there is a consistent decrease in β -sheet content with increasing assembly durations even we varied fitting parameters.

In Figure 3, we utilize the fitting parameters for silk fibroin as reported in the literature (i.e., Nature Communications, 2016, 7, 12334; Critical Reviews in Biochemistry and Molecular Biology, 1995, 30, 95-120; Acta Biomaterialia, 2018, 73, 355-364) to calculate the secondary structure content for doing qualitative analysis. In statistics, we calculated the average values and standard errors of the fitting results from 3-4 spectra which were repeatedly measured for each sample, as shown in Figure 3b-d and Figure S8-13.

Furthermore, to verify the consistency of the trend in secondary structure content changes, we varied the fitting parameters, including the baseline, number of peaks, and full width at half maximum (FWHM), to generate various fitting results for comparing the secondary structures of the proteins. These fitting results were then subjected to standard statistical analysis. As supplemented in Figure S14, there is a consistent decrease in β -sheet content with increasing assembly durations.

Changes made in the revised manuscript:

We thank the reviewer for this question, in response to which we have made the following changes in Lines 188-200 on Page 7 of the revised manuscript:

“As shown in Fig. 3b, we can extract the **average** content of secondary structures **with standard error for semi-quantitative analysis**: β -sheet is $\sim 79\% \pm 4\%$, random coil is $\sim 14\% \pm 3\%$, and turn is $\sim 7\% \pm 2\%$ at 15 min; β -sheet is $\sim 57\% \pm 4\%$, random coil is $\sim 40\% \pm 5\%$, and turn is $\sim 3\% \pm 1\%$ at 60 min; β -sheet is $\sim 49\% \pm 3\%$, random coil is $\sim 41\% \pm 2\%$, and turn is $\sim 10\% \pm 1\%$ at 300 min. The average secondary structure content, along with the standard error, are determined by applying the same fitting method to 3-4 extinction spectra (which were repeatedly measured for each sample, further details in Note 2).^{58, 59} These results are depicted in Figures 3b-d and Figures S8-13. To mitigate any potential errors in secondary structure content that could arise from variations in fitting parameters (such as baselines, number of peaks, and the full width at half maximum of deconvolution peaks), we generated a range of fitting results. As shown in Figure S14, there is a consistent decrease in β -sheet content with increasing assembly durations.”

Supplementary Figure S14. The change of β -sheet content at different assembly durations using different fitting parameters. (a)-(c) 279 K. (d)-(f) 299 K. (g)-(i) 348 K. It is noted that the secondary structure content is varied because different fitting parameters are utilized, such as different baselines, number and full width at half maximum (FWHM) of deconvolution peaks. By summarizing the fitting results with average value with standard error, we observed that the β -sheet content consistently decreases with increased assembly durations.

(4) Lines 187-190: “the oligomers with a height of 0.8 nm and 1.2 nm [...] have a high β -sheet content of \sim 79% and \sim 67%, respectively. This indicates that the oligomers are the assembly intermediates rather than SFs”: What is the basis for this statement? Can the authors explain in more detail why a higher β -sheet content is indicative of those structures being intermediates? How much higher are those β -sheet contents, compared to amorphous SF?

Reply: Changes in β -sheet content are important indicators of changes in protein secondary structure, which are indicative of assembly for silk fibroin (Nature Communications, 2021, 12, 3711; Science Advances, 2020, 6, eabb6030; Biophysical Chemistry, 2001, 89, 25-34). Whereas our oligomers have similar heights with a silk fibroin monomer (Nature Communications, 2022, 13, 7856), they have much higher β -sheet content ($>$ 50%) versus that of silk fibroin monomers in solution (\sim 9%) as depicted in Figure 3 and Figure S15. Thus, these oligomers are identified to be assembly intermediates rather than mere silk fibroin monomers.

Inspired by the reviewer's question, we have made corresponding modifications to the main text of our manuscript.

Changes made in the revised manuscript:

We thank the reviewer for this question, in response to which we have made the following changes in Lines 205-207 on Page 7-8 of the revised manuscript:

“Remarkably, the oligomers with a height of 0.8 nm and 1.2 nm (i.e., after assembling for 15 min at 279 K and 299 K, corresponding to Fig. 2a and Fig. 2b) have a high β -sheet content of ~79% and ~67%, which is much higher than the β -sheet content of SF (details in Fig. S15). Since secondary structure changes are a sign that assembly has occurred, the observed oligomers are assembly intermediates rather than SFs.”

(5) Line 202: β -sheets in silk materials are stable and durable structures (materials with high β -sheet contents can last for extremely long periods). However, the authors report a decrease in β -sheets during the SNF formation. Do the authors have any explanation for how the β -sheets present in oligomers might be disrupted, despite the high stability of such structures?

Reply: We thank the reviewer for the kind reminder. β -sheets are stable and durable below 120°C, as reported in Biochemistry 2014, 53, 6252–6263. Given that the maximum temperature in our experiments was 348 K (75°C), it is unlikely that β -sheets were disassembled in our study. Since the β -sheet content of well-assembled SNFs at the graphene interface (300 min, 348 K) is comparable to those assembled in heated solutions, as detailed in Note 3 and Fig. S16 of the Supplementary Information. The observed gradual reduction in the average β -sheet content during silk nanofibril assembly (Figure 3) is attributed to the diminished template effect of graphene, rather than disruption of β -sheets, as stated in Lines 196-200. “*As the assembly duration increases (from 15 min to 60 min and 300 min), mainly corresponding to the growth stage of SNFs, there is a gradual reduction in the template effect of graphene since it has been fully covered. In this process, there is a rapid decrease followed by stabilization in the β -sheet content.*” in the original manuscript. We have rewritten this part to provide a clearer explanation of this process in the revised manuscript (Lines 220-231 on Page 8, and Figure S17), as detailed below.

To reconcile the differences between simulation and experimental temperatures, we adjust the temperature parameters ($0.8 \cdot T$, T) in our coarse-grained simulations, ensuring no disruption of β -sheets and greater alignment with experimental data. These modifications have been incorporated into our revised manuscript (Pages 9-10, Lines 257-288), as detailed below.

Changes made in the revised manuscript:

We thank the reviewer for this question, in response to which we have made the following changes in Figure 4c, and Lines 215-226 on Page 7-8, and Lines 257-288 on Page 9-10, and Figure S18 in the revised manuscript and supplementary information:

“

Fig. 4 Multi-scale MD simulations of SNF assembly. **a** SF fragment model utilized in MD simulations. **b** Snapshots of SF fragment dynamics on graphene at nucleation stage (i.e., 0 μ s, 0.25 μ s, 0.5 μ s, and 1 μ s). The environment temperature is 343 K. The yellow, indigo, blue, and white parts represent β -sheet, turn, helix, and coil, respectively. **c** Snapshots of molecular interaction at the growth stage. Two environment temperatures (i.e., $0.8 \cdot T$ and T) are set. N is the average contact number.”

Lines 215-226 on Page 8 “As the assembly duration increases, the thickness of the assembly intermediates also increases, as demonstrated in Fig. 2d. This indicates that the assembly interface moves away from the graphene surface, leading to a reduced template effect and a consequent gradual decrease in the average β -sheet content. Furthermore, Fig. 3c and Fig. 2d (from 60 min to 300 min) illustrate that when the assembly interface is significantly distanced from graphene, the β -sheet content stabilizes, despite the increasing length and thickness of SNFs. It is also noteworthy that the β -sheet content of well-assembled SNFs at the

graphene interface (300 min, 348 K) is comparable to those assembled in heated solutions, as detailed in Note 3 and Fig. S16 of the Supplementary Information. Consequently, graphene plasmonic infrared sensor has minimal influence on the growth stage of SNFs.”

Lines 257-288 on Page 9-10 “Two models are built to clarify the impacts by temperature and graphene-induced oligomers during the growth stage. Since first layer of oligomers covered the graphene surface, the first model starts with four graphene-template-induced oligomers (blue, β -sheet content $\sim 51.2\%$), followed by introducing three new SF molecules (red), as shown in Fig. 4c. The second model begins with seven SF molecules to represent the assembly process without graphene, representing conditions similar to those in an aqueous solution, as shown in Fig. S18a.

To investigate the temperature effect on facilitating SNF elongation, we compare the average contact number (N) based on first model at varied temperature. As shown in Fig. 4c, the red molecules form well-ordered structure at 0.8^*T . Some red molecules tend to establish interfacial contacts along the geometric surface of their respective structures, resulting in a clearly defined contact interface (as magnified by a yellow dashed circle). It is noted that some red molecules exhibit no contact (as magnified by a black dashed circle), which disrupts the assembly elongation process and results in shorter SNFs. Thus, red assemblies have limited contacts ($N=521\pm 12$) with blue oligomers. In sharp contrast, when the temperature is increased to T , adjacent blue oligomers and red molecules undergo contact fusion promoted by random structures (as magnified by a green dashed circle). This results in an interconnected β -sheet network with numerous interfacial contacts ($N=1175\pm 15$), which provides the necessary stability for SNF elongation. Therefore, the simulation results are consistent with experimental observations that lower temperatures are unbeneficial for SNF elongation.

Furthermore, by summarizing β -sheet content in these two models, as shown in Fig. S18b, it is evident that the average β -sheet content experiences a decrease after the assembly process, compared with the initial graphene-induced oligomers (51.2%) at both temperatures. For instance, at 0.8^*T , the average β -sheet content drops to approximately $49.2\pm 0.3\%$. The reduction can be attributed to the newly added SF molecules (colored red) forming a β -sheet content of around 46.5% post-assembly, a structure closely matching that of the assembly formed at 0.8^*T in model 2 without graphene (β -sheet $\sim 46.4\pm 0.3\%$, as illustrated in Fig. S18). The observation underscores that the decrease in β -sheet content is a consequence of the diminishing template effect of graphene, aligning well with the experimental findings.”

(6) Furthermore, the authors use MD simulations to claim that SF in contact with graphene forms more β -sheets than in water. 1. However, I doubt that the MD simulations reported here are not likely to show the formation of β -sheets in water-only systems. 2. The timescale for the formation of β -sheets is unlikely to be captured by MD simulations, especially if no replica-exchange sampling is used. 3. I think that the simulations reported here can only be used to demonstrate that SF has a high tendency to adsorb on graphene.

Reply: We thank the reviewer for raising these questions. As detailed in the response to question (1) from Reviewer 2, graphene facilitates the nucleation of β -sheets and does not disrupt the β -sheet content of SNFs that are now in direct contact with it. Acknowledging the challenge in capturing the timescale of β -sheet formation in water-only systems using all-atom molecular dynamics (MD) simulations, our comparison is limited to 1 μ s results. Here, we only utilized MD simulations to claim that SF in contact with graphene **accelerates the formation β -sheet** rather than forms more β -sheets than in water. We agree with that our simulation results just show that the graphene accelerate the formation of β -sheets, as we claimed that the graphene leads to accelerated formation of β -sheets compared to the water-only system. This is indicated in Lines 255-257 “*Furthermore, a comparison of the assembly behavior of SF with and without graphene reveals that graphene can indeed accelerate the rate of SF conformational transitions.*” of the original manuscript. To prevent any misunderstandings regarding the increased β -sheet content due to graphene, we have revised the label and figure caption of Fig. S17b and c.

Changes made in the revised manuscript:

We thank the reviewer for this question, in response to which we have made the following changes in Figure S17 of the revised supplementary information:

Supplementary Figure S17. The nucleation stage of SNF assembly on graphene. (a) The flow of all-atom MD simulation. (b) The **average number with standard error** of hydrogen bonds inside SF and between SF and water at 343 K **after 1 μ s assembly**. (c) Simulated **average β -sheet content with standard error** of SF with or without graphene at 343 K **after 1 μ s assembly**. **It is important to underline that, within this 1 μ s timeframe, conformational changes do not fully occur in either the systems with or without graphene. The purpose of comparing SF with and without graphene at 1 μ s is to demonstrate that graphene can accelerate the assembly rate of SF. This finding suggests a templating effect of graphene in facilitating the conformational transition of SF.** (d) Snapshots of SF assembly on graphene surface at different temperatures (283 K, 303 K, 323 K, 343 K, and 363 K) with different moments (0.25 μ s, 0.5 μ s, 0.75 μ s and 1 μ s)..

(7) Discussion: The authors indicate that SF is already stored in the glands of silkworms as a

solution of oligomers rich in β -sheets that is subsequently spun into fibers. However, the current understanding of this process is that spinning induces the formation of even more β -sheets due to the alignment of fibers. In contrast, this paper shows a decrease in β -sheets. I am therefore not sure how the results here reported could help in elucidating the formation of SNFs in Nature, which is the main aim of this paper as indicated by the authors. Could the reduction in β -sheets be caused by the hydrophobicity of graphene?

Reply: We agree with the reviewer that the reduction in β -sheets is caused by the graphene, i.e., a reduced template effect of graphene on silk nanofibril assembly as their thickness increases. This explanation is provided in greater detail in reply to question (5). Consequently, our findings do not contradict the formation of silk nanofibril (SNF) in Nature in terms of β -sheet content. Furthermore, our identification of β -sheet-enriched oligomers present prior to SNF formation aligns with experimental observations of β -sheet-enriched nanocompartments in silk gland (Nature Communications, 2022, 13, 7856). It is worth noting that this assembly process has never been explored previously due to the lack of characterization methods. For the first time, we demonstrate the potential of utilizing graphene plasmonic infrared sensor to investigate the SNF assembly process by connecting their conformational and morphological transitions. Our current understanding of silk nanofibril assembly has provided new insights into its storage mechanisms as stated in Lines 307-317 of the revised manuscript, highlighting the significance of β -sheet-dominated oligomers and short-rod-like silk nanofibrils. These elements might be crucial for the rapid consolidation of silk, given the constraints of limited spinning time and external stimuli.

Changes made in the revised manuscript:

We thank the reviewer for this question, in response to which we have made the following changes in Lines 307-310 on Page 11 of the revised manuscript:

“ Particularly, the discovery of β -sheet-enriched oligomers present prior to SNF formation aligns with the previous experimental observations of β -sheet-enriched nanocompartments in silk gland⁶², which motivates us to speculate about a potential pathway in the silk spinning process:”

(8) Lines 301-302: “To obtain a 9.3 mol/L SF solution, degummed silk at a concentration of 10% w/v was dissolved in a LiBr solution”. Can the authors confirm that this is the method that they used? Based on my experience on common silk degumming methods, the 9.3 M solution would be from LiBr, not from SF. I believe there is an error in this sentence.

Reply: We thank the reviewer for the kind reminder. We apologize for the mistake on the method description. In fact, we dissolve the degummed silk with a concentration of 10% w/v in a 9.3 mol/L LiBr solution (Nature Protocol, 2011, 6, 1612–1631). We have corrected the description in the revised manuscript.

Changes made in the revised manuscript:

We thank the reviewer for this question, in response to which we have made the following

changes in Line 339 on Page 12 of the revised manuscript:

“The degummed silk fiber at a concentration of 10% w/v was dissolved in a 9.3 mol/L LiBr solution with a water bath at 333 K for 2 hours.”.

(9) Line 329-331: “the sensor was cleaned thoroughly by rinsing it with deionized water to remove any loosely adsorbed assembly intermediates. The sensor was dried by gently blowing nitrogen gas to ensure no residual moisture remained”. Since SF can hydrophobically adsorb on graphene, would it be possible for the authors to provide evidence that this cleaning procedure is sufficient to remove adsorbed SF from the graphene surface?

Reply: We thank the reviewer for raising this question. As shown in Fig. 4b, silk fibroin experiences a conformational change upon adsorption onto graphene, marking the start of the assembly process. Once adsorbed, this silk fibroin is notably difficult to detach from the graphene surface. In our experimental protocol, the sensor is cleaned using deionized water. This step is aimed primarily at reducing silk fibroin deposition on the substrate caused by water evaporation from the silk fibroin solution, rather than removing the adsorbed silk fibroin itself. Additionally, this cleaning process is effective in eliminating loosely adsorbed assembly intermediates. The success of our cleaning method is apparent in the morphology images presented in Figure 2a-c. These images show very few irregular silk fibroin aggregates on graphene, confirming the graphene surface's cleanliness after the cleaning procedure.

Changes made in the revised manuscript:

We thank the reviewer for this question, in response to which we have made the following changes in Lines 366-370 on Page 13 of the revised manuscript:

“At specific assembly durations, the sensor was cleaned thoroughly by rinsing it with deionized water to avoid protein deposition on the substrate because of water evaporation in the residual SF solutions. The sensor was dried by gently blowing nitrogen gas to avoid any potential damage or contamination.”

(10) Note 4, SI: at what distance do the authors place the SF polypeptide on top of the graphene surface for MD simulations?

Reply: In the initial configuration in the MD simulation, we placed silk fibroin ~1 nm on the graphene surface and ensured that there was a water layer interval between silk fibroin and graphene, as shown in Figure R1.

Figure R1 Measured distance between silk fibroin and graphene in MD simulation.

9. Some minor typos:

- Line 108: “excite”, not “excited”.
- Line 131: “involves”, not “involve”.
- Line 246: “it exhibits” instead of “there exhibits”.

Reply: We thank the reviewer for pointing out these grammar issues. These words have been corrected in the revised manuscript.

Changes made in the revised manuscript:

We thank the reviewer for this question, in response to which we have made the following changes in Line 112 on Page 4, Line 140 on Page 5 of the revised manuscript:

Line 112 on Page 4 “Meanwhile, the graphene nanoribbon widths were designed and V_G is tuned to excite graphene plasmons in proper frequency range (i.e., 1500-1700 cm^{-1}).”

Line 136 on Page 5 “We propose that the assembly process of SNFs involves nucleation, growth, and a final plateau.”

Reviewer #3 (Remarks to the Author):

This manuscript reports in situ measurement of silk nanofibril assembly using graphene nanoribbon-based FTIR sensing. The authors prepared silk fibroin from silkworm cocoons and assembled silk fibroin to silk nanofibrils on a patterned graphene substrate, which is also used as a FTIR sensing substrate, under different temperature and duration to understand silk nanofibril assembly mechanisms. Surface morphologies and FTIR spectra in the process of nanofibril assembly are mainly discussed, and molecular dynamics simulations were used to support experimental results. The reviewer has concerns regarding materials characterizations and design of FTIR (see comments 2, 3) and the clarity in the discussions about the novelty of the work (see comments 1, 4).

1. The reviewer thinks the title does not represent the work reported here because it is not specific enough. The authors may want to consider changing the title.

Reply: We thank the reviewer for the suggestion. To represent this work specifically, we have changed the title of the manuscript to “In-situ Observation of Silk Nanofibril Assembly

via Graphene plasmonic infrared sensor”.

2. The authors used patterned graphene as a substrate to enhance the silk nanofibrils assembly as well as for graphene plasmon-enhanced FTIR.

2-1. Graphene plasmon-enhanced FTIR: plasmonic properties of graphene are highly depending on the geometry and length scale of graphene patterns. The authors mentioned that “the graphene nanoribbon widths were designed to excited graphene plasmons in proper frequency range,” in the Results section, but any further details are missing. Based on the AFM images in Figure 2, the width of the nanoribbon can be roughly guessed, but the reviewer believes that more details should be provided. Specifically, why did the authors decide to have roughly 75 nm for the width and 75 nm for the distance between ribbons? What are corresponding plasmonic properties/effects from such patterns? Please add more details with relevant references and/or simulation results to support the structure used in this work.

Reply: We thank the reviewer for the suggestion. To excite graphene plasmon with a proper resonance frequency (1500-1700 cm^{-1}) and highest extinction, we designed graphene nanoribbons with width of 50-90 nm and with period-to-width ratio $\approx 2:1$ according to its plasmonic properties, as reported in literatures:

The dispersion of graphene plasmon is $\omega_{pl} = \sqrt{\frac{e^2 |E_F| q}{2\pi \hbar^2 \epsilon_0 \epsilon_r}}$, where ϵ_0 is the dielectric constant of air, ϵ_r is the average dielectric constant of its surrounding medium, E_F is the fermi energy of graphene, q is the wave vector and $q = \pi/W$ (W is the width of graphene nanoribbon). As shown in Figure R2, we can find that the resonance frequency of graphene plasmon (ω_{pl}) is related to W and E_F of graphene nanoribbon (Nature Photonics, 2013, 7, 394-399; ACS Nano, 2014, 8, 1086–1101.). In addition, when the width of a graphene nanoribbon is designed to be comparable to the spacing between them, the loss by plasmon–plasmon interactions between adjacent nanoribbons is minimized. Thus, the extinction of graphene plasmon is optimized. (ACS Photonics 2018, 5, 9, 3459–3465.) By following the modulation mechanism of graphene plasmon, the graphene nanoribbon widths were carefully designed and V_G is tuned to excite graphene plasmons in proper frequency range (i.e., 1500-1700 cm^{-1}).

Figure R2. (a) The extinction spectra of graphene plasmon with varying nanoribbon width, and (b) varying doping of graphene (reused from Nature Photonics 2013, 7(5): 394-399.).

Changes made in the revised manuscript:

We thank the reviewer for this question, in response to which we have made the following changes in Lines 112-118 on Page 5 of the revised manuscript:

“Meanwhile, the graphene nanoribbon widths were designed and V_G is tuned to excite graphene plasmons in proper frequency range (i.e., 1500-1700 cm^{-1}). This is because the resonance frequency of graphene plasmon is related to width and Fermi energy of graphene nanoribbon according to the dispersion^{52, 53}. In addition, when the width of a graphene nanoribbon is designed to be comparable to the spacing between them, the extinction of graphene plasmon is optimized and the plasmon–plasmon interactions between adjacent nanoribbons is minimized⁵⁶.”

2-2. Material: The authors specified that they purchased graphene from a vendor. Was the graphene monolayer or thicker? Either way, please provide materials characterization results as the number of layers of 2D materials has a significant impact on the properties, the reviewer believes that material characterization results should be presented even though it was a commercially available product.

Reply: We appreciate the question posed by the reviewer. To substantiate the monolayer property and quality of the commercial graphene used in our study, we performed optical microscopy and Raman spectroscopy characterizations. Further information can be found in Figure S2. Optical microscopy, as shown in Figure S2a, reveals minimal contrast variation in the graphene regions, suggesting a consistent layer throughout. Following this, Raman spectroscopy was employed. As depicted in Figure S2b, the characteristic G' peak of graphene appears at 2692 cm^{-1} , with the G peak at 1593 cm^{-1} . Notably, the G' peak exhibits a distinct single Gaussian profile, with an intensity about twice that of the G peak, confirming the monolayer status of our graphene.

Changes made in the revised manuscript:

We thank the reviewer for this question, in response to which we have made the following changes in Lines 100-101 on Page 4 of the revised manuscript, and supplement optical microscopy and Raman spectroscopy characterizations in Figure S2 of revised Supplementary information:

“As shown in the right panel of Fig. 1b, a typical AFM image displays the SNFs selectively assembled on the graphene nanoribbons with a total height of approximately 7.0 nm (i.e., includes ~5.0 nm-thick SNFs and ~2.0 nm-thick monolayer graphene) after 300 minutes of assembly (details in Fig. S2 and Fig. S3).”

“

Supplementary Figure S2 Characterization of graphene plasmonic infrared sensor. (a) Optical microscope image of graphene film and graphene nanoribbon region. (b) Raman spectrum of utilized graphene film. 514 nm laser was used. The optical and Raman data show the graphene film is continuous and monolayer.”

2-3 Electrical characterization of graphene plasmonic infrared sensor: In the subsection of “Characterization of the graphene plasmonic infrared sensor” of the Methods section, the authors discussed: “The electrical properties of the graphene plasmonic infrared sensor were characterized using a source meter (Keithley 2636B). The source meter allows for the evaluation of electrical performance of every sensor. Additionally, the source meter was used to tune the back gate voltage, enabling control of electrical properties and modulation of the graphene plasmonic response.” However, there is no data or discussion related to these electrical characterizations in both manuscript and supplementary information. Additionally, if the authors tuned graphene’s plasmonic response by applying electric fields, the specific conditions should be discussed somewhere in the manuscript as well.

Reply: Thanks to the reviewer for the suggestions. We have supplemented the discussion in the revised manuscript Lines 108-111. The data of electrical results is supplemented in the revised supplemental material as shown in Figure S4.

Changes made in the revised manuscript:

We thank the reviewer for this question, in response to which we have made the following changes in Lines 108-111 on Page 4 of the revised manuscript:

“The graphene plasmonic infrared sensor was fabricated as a field-effect transistor as shown in Fig. 1b. Thus we can measure the electrical transfer characteristic curve (V_G - I_{SD}) to ensure the adjustability of graphene Fermi level and extract the CNP of graphene⁵³ as demonstrated in Methods and Fig. S4.”

Figure S4 is supplemented in the revised supplementary information:

Supplementary Figure S4 Electrical performance of graphene plasmonic infrared sensors.

(a) The V_{SD} - I_{SD} data, and (b) V_G - I_{SD} data of the graphene plasmonic sensor (Device 1), as utilized in Fig. 1c. (c) The V_{SD} - I_{SD} data, and (d) V_G - I_{SD} data of the graphene plasmonic sensor (Device 2), as utilized in Fig. 2a&Fig. 3b. (e) The V_{SD} - I_{SD} data, and (f) V_G - I_{SD} data of the graphene plasmonic sensor (Device 3), as utilized in Fig. 2b&Fig. 3c. (g) The V_{SD} - I_{SD} data, and (h) V_G - I_{SD} data of the graphene plasmonic sensor (Device 4), as utilized in Fig. 2c&Fig. 3d. V_{SD} and I_{SD} represent the applied voltage and the measured current between the source and drain, respectively. V_G refers to the applied gate voltage. V_{SD} - I_{SD} curves were recorded with V_G set to 0 V, while V_G - I_{SD} curves were obtained with V_{SD} at 0.1 V. The linear slope observed in the V_{SD} - I_{SD} curve suggests good contact between the graphene and the electrodes. The lowest point on the V_G - I_{SD} curve indicates the charge neutrality point (CNP). Adjusting V_G can modify the Fermi level; for instance, in Fig. S4h, a V_G lower than the CNP (indicated by the red shaded area) shifts the Fermi level towards the valence band, indicating p-doping of graphene.

3. Figure 2d: Figure 2d presents the height change of the silk nanofibril structures formed on graphene nanoribbons. The reviewer was not totally convinced that long-rod-like silk nanofibrils would have a height higher than oligomers and short-rod-like silk nanofibrils.

Longer fibrils may be stacked more efficiently and thus may have a lower height, compared to oligomers and short-rod-like fibrils, as described in the small schematic illustrations inserted in the panel d. The reviewer suggests that the authors discuss a bit more details about the height comparisons for future readers.

Reply: We thank the reviewer for the suggestions. We agree with the reviewer that the measured height of the silk nanofibrils formed on the graphene nanoribbons indeed represents the height of multiple nanofibrils stacked together, as shown in Fig. 2d. We want to stress that in our study the overall height of stacked proteins is most critical for analyzing their secondary structure at different assembly stages. Silk fibroin underwent a conformational transition on the graphene surface, leading to the formation of oligomers and then silk nanofibrils. We observed that the 0.8 nm-thick oligomers initially covered on the graphene surface had a significantly higher β -sheet content (~79%) compared to the thermal induced silk nanofibrils (~58%) formed in aqueous solution. Both the experimental and theoretical results suggest that graphene has a template effect for the silk fibroin assembly. As the assembly duration increases, the thickness of the assembly intermediates also increases, as demonstrated in Fig. 2d. This indicates that the assembly interface moves away from the graphene surface, leading to a reduced template effect and a consequent gradual decrease in the average β -sheet content.

Changes made in the revised manuscript:

We thank the reviewer for this question, in response to which we have made the following changes in Lines 158-160 on Page 6 and Lines 215-226 on Page 8 of the revised manuscript:

Lines 160-162 on Page 6 “Here, the height of the SNFs formed on the graphene nanoribbons we measured represents the height of multiple nanofibrils stacked together.”

Lines 215-226 on Page 8 “As the assembly duration increases, the thickness of the assembly intermediates also increases, as demonstrated in Fig. 2d...Consequently, the graphene plasmonic infrared sensor has minimal influence on the growth stage of SNFs.”

4. The reviewer was curious about the new insight the authors revealed from this work. Specifically, was this the first paper reporting in situ measurement of silk nanofibril assembly? Was this the first paper using a graphene plasmon-enhanced FTIR sensor to study silk nanofibril assembly? Was this the first paper having silk nanofibril assembly on a graphene substrate? The authors mentioned in the abstract that “this work’s results offer a detailed understanding of the silk nanofibril’s assembly mechanism,” but it is not clear which part of the mechanism is newly reported. Please discuss the novelty of your work clearly to help your readers understand your work better.

Reply: We thank the reviewer for raising these important questions. The novelty of our work lies in the first-time in-situ monitoring of both the secondary structure and morphological changes during silk nanofibril assembly simultaneously. Previously, these two aspects have never been directly correlated due to the lack of a suitable method for characterizing assembled

silk nanofibrils with thicknesses less than 10 nm, as discussed in the Introduction (paragraph 3) and shown in Table R1. While the morphology of silk nanofibrils on graphene has been previously explored (ACS Macro Letters 2014, 3, 146–152), our study is the first to utilize a graphene plasmonic infrared sensor to investigate silk nanofibril assembly.

For the first time, our study links morphology and secondary structure data over extended molecular timeframes and varied temperatures. We reveal that amorphous silk fibroins undergo a conformational transition to β -sheet-rich oligomers, which subsequently connect to form silk nanofibrils. This discovery fills a critical gap in understanding the transformation from silk fibroin to silk nanofibril, the fundamental building blocks of silk fibers, which endow them with exceptional properties.

Table R1. Experimental approaches of in-situ studying silk nanofibril assembly

Methods	Sample preparation requirements	Nanoscale morphology	Conformation	Ref.
Small Angle X-ray Scattering	Homogeneous sample in solution or non-crystalline states	×	√	Adv. Mater. Tech. 2021, 6(10): 2100124.
Nuclear magnetic resonance spectroscopy	Purity sample >95%	×	√	Nature 2010, 465, 239.
Circular dichroism spectroscopy	Homogeneous sample in solution with a concentration (0.01-0.2 g/L)	×	√	Biopolymers 2014, 101, 1181-1192.
Fluorescence	Sample with fluorescent labels	×	√	Biomacromolecules 2006, 17(11): 3570-3.
Raman spectroscopy	Sample with low fluorescence	×	√	Sci. Adv. 2020, 6(45): eabb6030.
Sum-frequency generation spectroscopy	Sample on liquid-gas or liquid-solid interfaces	×	√	Langmuir 2018, 34(32): 9453.
Fourier-transform infrared spectroscopy	Sample in powder or film with thickness >1 micrometer	×	√	ACS BIOMATER SCI ENG 2016, 2, 1298.

Sample in solution concentration: with a concentration (0.1-10 g/L)				
Our approach*	Sample with thickness>0.8 nm	✓	✓	/

Changes made in the revised manuscript:

We thank the reviewer for this question, in response to which we have made the following changes in Lines 29-35 on Page 1-2 and Line 296-313 on Page 10-11 of the revised manuscript: Lines 30-34 on Page 1 “This approach uniquely reveals the secondary structure of nanoscale assembly intermediates (0.8 – 6.2 nm) and their morphological evolution. It also provides insights into the dynamics of SF over extended molecular timeframes. Our novel findings reveal that amorphous SFs undergo a conformational transition towards β -sheet-rich oligomers on graphene. These oligomers then connect to evolve into SNFs. These insights provide a comprehensive picture of SNF assembly, paving the way for advancements in biomimetic silk spinning.”

Lines 293-310 on Page 10-11 “In summary, we introduce an innovative experimental approach utilizing graphene plasmonic infrared sensor, which can identify the secondary structure contents and corresponding morphologies of assembly intermediates (ranging from 0.8 nm to 6.2 nm in thickness) during SNF assembly. By combining multi-scale MD simulations, we gain insights into the molecular mechanisms underlying protein-protein and protein-environment interactions. A SNF assembly model is thus provided as illustrated in Fig. 1a. In nucleation stage,... which motivates us to speculate about a potential pathway in the silk spinning process:”

5. Nanofibril assembly temperature: The nanofibril assembly process was measured in different temperatures including 279 K, 299 K, and 348 K. 348 K showed the fastest assembly. As 348 K seems very high for a living organism, the reviewer was wondering at which temperature a silkworm produces its silk.

Reply: We thank the reviewer for raising the question. Silkworms, particularly *Bombyx mori*, produce silk in temperatures ranging from 298 K to 303 K. This range is conducive to their living and feeding conditions (Biomacromolecules, 2007, 8, 1, 175–181). Natural silk production, or silk spinning, occurs without the need for external heating or temperature control. In contrast, artificial silk spinning often involves higher temperatures (Adv. Healthcare Mater., 2020, 9, 1901552). Temperature plays a crucial role in determining the molecular alignment and crystallinity of the spun fibers. Therefore, understanding the effects of temperature on silk spinning is key to optimizing artificial silk production, improving fiber properties.

6. The authors use the term “assembly velocity” multiple times in the manuscript. However, it

sounds strange to me, as velocity is a vector property. Could you use “assembly rate” or another term?

Reply: We thank the reviewers for their suggestions. We have used “assembly rate” to replace “assembly velocity” in the revised manuscript.

Changes made in the revised manuscript:

We thank the reviewer for this question, in response to which we have made the following changes in Line 155 and 158 of the revised manuscript:

“To compare the assembly **rate** at different durations under varied temperatures, we summarized the heights of oligomers and SNFs in Fig. 2d (details in Fig. S2). As the temperature rises, the thicknesses of the short rod-like SNFs on graphene also increase, and the lengths of the SNFs become longer, indicating a higher assembly **rate**.”

Reviewer #4 (Remarks to the Author):

The manuscript by Wu et al. studies the assembly process of silk nanofibrils (SNFs) using graphene ribbon plasmonic infrared sensor. They monitor conformational transitions of silk fibroin to SNF and proposes an assembly model. Then the authors discuss the potential implications for artificial silk spinning techniques.

While the study presents some interesting understanding of the assembly mechanism of SNFs, this reviewer finds that novelty is lacking to publish this in Nature Communications. The research uses graphene ribbon system, which is by now well-established for sensing applications. Specifically, their focus on the graphene nanoribbon sensor's application in silk fibril assembly in a dry environment is too narrow in scope, not generally applicable to a wide range of proteins, and is not novel any more.

Finally, the implications of the study, primarily centered on silk spinning techniques, may not resonate strongly with the broad readership of Nature Communications.

I suggest that the authors submit this manuscript to a more specialized journal.

Reply: We thank the reviewer for the comments. Our study is pioneering in its use of a graphene plasmonic infrared sensor to investigate silk nanofibril assembly. Although graphene plasmon-enhanced infrared spectroscopy has been widely applied on characterizing fingerprints of protein molecules, it has never been applied to probe protein assembly process and monitor its conformational changes. This is because the plasmonic hotspot region can hardly align with the assembly interface of protein. However, we discovered that graphene not only facilitates silk nanofibril assembly but also anchors assembly intermediates, making graphene plasmons an ideal platform for investigating assembly mechanism of silk nanofibril.

Furthermore, we want to highlight the broad readership for our work. Silk fibers, as a natural protein, hold great potential for a wide range of applications in flexible brain–computer interfaces, tissue engineering, and wearable textiles. Therefore, this field attracted widespread attention and research interests (e.g., *Nature Materials*, 2010, 9, 511-517; *Proceedings of the National Academy of Sciences of the United States of America*, 2012, 109, 7699-7704; *Nature Reviews Chemistry*, 2023, 7, 302-318). In natural silk spinning, the silk proteins are synthesized and stored in silk glands at high concentrations. Upon spinning, the dope is forced to flow through a specially shaped spinning duct and is subjected to a series of physiological changes such as shear, pH, metal gradient, and salt gradients. During the process, the silk proteins undergo phase transitions along with self-assembly to form solid, semi-crystalline, insoluble fibers (*Applied Physics Reviews*, 2020, 7, 011313). Deciphering the silk spinning mechanism of silkworms and understanding the relationship between silk fiber structure and function is a pivotal step in the fabrication of high-performance fibers. Thereinto, studying the formation mechanism of silk nanofibril is of paramount importance as a fundamental block of silk fibers (*Nature Reviews Materials*, 2018, 3, 18016; *Nature Materials*, 2010, 9, 359-367; *Science Advances*, 2020, 6, eabb6030; *Nature Communications*, 2021, 12, 3711.). Nevertheless, the secondary structure and morphology information have never been directly linked together is due to lack of a proper method to characterize both for the assembled silk nanofibrils with thickness of less than 10 nm (as shown in Table R1 replied in question (4) of Reviewer 3, and discussed in manuscript paragraph 3 of the Introduction,).

For the first time, we achieve simultaneous in-situ monitoring of the secondary structure and morphological changes during the assembly of silk nanofibrils. The correlation of morphology and secondary structure data across extended molecular timeframes and under varying temperatures has led to a significant discovery: amorphous silk fibroins undergo a conformational transition towards β -sheet-rich oligomers, which subsequently link to form silk nanofibrils. This insight fills a critical knowledge gap regarding the transformation from silk fibroin to silk nanofibril, the foundational elements of silk fibers that endow them with their exceptional properties. We are confident that our findings will substantially contribute to the understanding of silk spinning mechanisms and inspire innovative approaches in artificial spinning strategies.

REVIEWER COMMENTS

Reviewer #1 (Remarks to the Author):

The authors have responded to my remarks in a mostly positive and convincing way. The model of oligomer assembly (Figure 1a) and contact fusion (Figure 4c) is well explained. This model applies to the specific situation of assembly by model silk proteins on a graphene surface. The authors suggest -speculatively- a possible link to natural silk spinning and mention the interesting observation of nanocompartments in the Anterior-Middle duct, where elongational flow is starting to induce fibrillation (Eliaz et al.,). I just want to point out that the β -sheet fraction in dragline-type silk fibers is only 10-15 % while the rest of the fiber is composed of disordered chains, forming various other structural motifs. A fibrillation model based principally on the β -sheet fraction is therefore highly questionable.

My main (minor) modification request concerns Figure 1a, which shows an oligomer bead with a folded, single β -sheet. The oligomer beads assemble into short-rod-like SNFs and subsequently into long-shod-like SNFs. The beads in the rods seem to have a preferred orientation -shown as lines or arrows- which is attributed to alignment and elongation in the flow. The authors should clarify and better depict how the β -sheet peptidic chain axis of the assembled beads (red lines) are oriented as I cannot deduce this from the comparison of the oligomer with the short-rod like SNFs. Are they parallel to the flow direction or normal to the flow direction? If the model would apply to biospinning one would expect alignment along the rod axis, in the same way as β -sheets are aligned along the silk fiber axis.

Reviewer #2 (Remarks to the Author):

I am pleased to confirm that I am satisfied with the changes made to the manuscript in its revised form. All my concerns have been properly addressed. As a result, I am happy to recommend this for publication.

Reviewer #3 (Remarks to the Author):

I appreciate the revised manuscript and the response from the authors to the reviewer's comments. Many of them are resolved, but some concerns persist. The numbering here refers to my previous review comments.

2-1. Graphene plasmon-enhanced FTIR: While the reviewer appreciates the response from the authors, the reviewer would suggest that the authors add some of those details to their manuscript. The reviewer overall feels that the important details of characterization conditions are still missing.

a) The graphene pattern design is very important to "observe" silk nanofibril assembly in the reported work, as the pattern design is directly related to the plasmonic properties of the "sensor." The reviewer still thinks sufficient experimental details are not provided. Specifically, what exactly was the width of the graphene ribbons? The authors mentioned that "we designed graphene nanoribbons with width of 50-90 nm," but it is not sufficient. What was the exact width the authors aimed for? As the authors mentioned the resolution of the E-beam lithography was 5 nm, the width range of 40 nm is not reasonable. In addition, the reviewer recommends that the authors add information about the width of graphene ribbons and the period-to-width ratio in the manuscript.

b) Please add the discussions related to the graphene design process (as the authors provided in the response) in the supplementary information or manuscript.

c) In the revised manuscript (LL112-118), the authors mentioned that "the graphene nanoribbon widths were designed and VG is tuned to excite..." So, how high (or low) was the gate voltage to

excite graphene plasmon in the nanoribbon? Did you use the same gate voltage for all measurements?

2-2. Materials

a) It is difficult to agree that the graphene used in this work is monolayer while the reviewer understands the authors' claim on the G' peak profile. First, the intensity ratio ($I_{G'}/I_G$) is clearly less than 2. Second, the AFM profile in Figure S3 does not reflect the claim. It is very common that the edge of patterned graphene shows a spike in AFM height profile, but the height difference between the center of the pattern and the substrate is clearly >1 nm, which implies that the pattern has 2 or more layers of graphene.

b) Optical microscopy image (Figure S2) could support the continuity and monolayer. However, the pattern here is too fine to show anything with an optical microscope. The reviewer was curious why the authors did not use scanning electron microscopy to show the patterns of the graphene layer.

Based on the revised manuscript, comments from all reviewers, and response to the review comments, I am not fully convinced that this manuscript is acceptable for publication in Nature Communications, considering the broad readership of the journal. I clearly see new findings and the reported work's contributions to the silk nanofibril research communities, so perhaps, it is more suitable to publish this manuscript in a more specialized journal.

RESPONSE TO REVIEWERS' COMMENTS

We appreciate the insightful and constructive comments raised by the reviewers. Here, we have addressed all the comments. Our responses are shown in blue, and our revisions to the manuscript and supplementary materials are indicated in red.

Reviewer #1 (Remarks to the Author):

(1) The authors have responded to my remarks in a mostly positive and convincing way. The model of oligomer assembly (Figure 1a) and contact fusion (Figure 4c) is well explained. This model applies to the specific situation of assembly by model silk proteins on a graphene surface. The authors suggest -speculatively- a possible link to natural silk spinning and mention the interesting observation of nanocompartments in the Anterior-Middle duct, where elongational flow is starting to induce fibrillation (Eliaz et al.). I just want to point out that the β -sheet fraction in dragline-type silk fibers is only 10-15 % while the rest of the fiber is composed of disordered chains, forming various other structural motifs. A fibrillation model based principally on the β -sheet fraction is therefore highly questionable.

Reply: We appreciate the reviewer's valuable feedback regarding the β -sheet crystalline content in dragline-type silk fibers. While studies report this content to be typically around 10-15% (e.g., *Macromolecules*, 1997, 30, 2860), we would like to respectfully propose two key considerations that support the validity of our fibrillation model based on the β -sheet fraction:

Firstly, it's important to distinguish between the total β -sheet content and the crystalline fraction detected by XRD. Techniques like FTIR, Raman, and NMR spectroscopy can reveal a higher total β -sheet content (encompassing both crystalline and non-crystalline regions) in *N. clavipes* dragline silk – often reaching 40-60% (Small 2019, 15, 1903948; *Chem. Commun.*, 2010, 46, 6714; *Biomacromolecules*, 2010, 11, 192; *Biophys. J.*, 2007, 92, 2885). This discrepancy arises because not all β -sheets possess the ordered structure necessary for XRD detection. Figure S1 further clarifies this point by showcasing the variations in β -sheet content and crystallinity across different silkworms and spiders, emphasizing β -sheet as a significant component in all of them.

While a complete picture of the silk fiber assembly mechanism remains a challenge, focusing on the changes in β -sheet content offers a valuable perspective. Extensive research has established the crucial role of β -sheets in determining the mechanical properties of natural silks (*Nat. Rev. Mater.*, 2018, 3, 18008; *Nat. Commun.*, 2020, 11, 1630; *Nano Lett.* 2023, 23, 3, 827; *Nature Mater.*, 2010, 9, 359; *Proc. Natl. Acad. Sci. USA*, 2002, 99: 6460). Therefore, tracking how β -sheet content evolves during the assembly process holds significant importance. Numerous studies have utilized these changes to gain insights into silk assembly mechanisms (*Appl. Phys. A*, 2006, 82, 223; *Biopolymers*, 2014, 101: 1181; *Nat. Commun.*, 2022, 13, 7856). Establishing a direct correlation between conformational data and nanoscale morphologies remains challenging due to the diverse sample preparation requirements for different techniques. To address this limitation, we present our innovative approach using a graphene plasmonic

infrared sensor. This sensor allows us to simultaneously identify the secondary structure content and corresponding morphologies of assembly intermediates. We have incorporated these points into the revised manuscript.

Sample name	Overall content of β -sheet [%]	Crystallinity [%]
B. mori cocoon silk	49	40
Spider dragline [10 mm s ⁻¹]	51	22
B. mori cocoon silk	54	41
A. pernyi silk	49	26

Figure R1: Summarization of β -sheets structure in silkworm and spider (Small 2019, 15, 1903948.).

Changes made in the revised manuscript:

We thank the reviewer for this question, in response to which we have made the following changes in Lines 177-183 on Page 6-7 and Lines 308-326 on Page 11 in the revised manuscript:

Lines 177-183 on Page 6-7: “Therefore, we evaluate the β -sheet content of both oligomers and SNFs in relation to their morphology using graphene plasmon-enhanced FTIR. **While comprehensively describing SNF assembly presents a significant challenge, focusing on changes in β -sheet content offers a valuable perspective for understanding the fibrillation process. This approach is supported by extensive research that has demonstrably linked β -sheets to the remarkable mechanical properties of natural silks^{8, 46, 47}.**”

Lines 308-326 on Page 11: “**The assembly process for regenerated *Bombyx mori* SF is thus provided as illustrated in Fig. 1a. In nucleation stage, unfolded SF molecules undergo a conformational transition to form β -sheet-enriched oligomers. These oligomers then progress to the growth stage, connecting with adjacent molecules and elongating into short-rod-like SNFs. The β -sheet peptide chain axis of the assembled oligomers has a preferred orientation – parallel to the elongation axis of the SNFs.** Ultimately, the short-rod-like SNFs align and elongate, becoming thicker and longer to form long-rod-like SNFs. Additionally, the critical roles of graphene-mediated interface and temperature in the SNF assembly process have been revealed. Specifically, the graphene interface accelerates SNF nucleation, while higher assembly temperatures enhance both SNF nucleation and elongation.

The discovery of β -sheet-enriched oligomers present prior to SNF formation aligns with the previous experimental observations of β -sheet-enriched nanocompartments in *Bombyx mori* silk gland⁵⁴, which motivates us to speculate about a potential pathway in the silk spinning process of *Bombyx mori*: During storage in the middle silk gland, SF has already assembled into β -sheet-rich oligomers and short-rod-like SNFs, which remain stable due to the presence of a hydrophilic coating. These β -sheet-dominated oligomers and short-rod-like SNFs can rapidly fuse into elongated nanofibrils **during spinning**. This is essential for the rapid

consolidation of silk under the constraints of limited spinning time and external stimuli.”

(2) My main (minor) modification request concerns Figure 1a, which shows an oligomer bead with a folded, single β -sheet. The oligomer beads assemble into short-rod-like SNFs and subsequently into long-shod-like SNFs. The beads in the rods seem to have a preferred orientation -shown as lines or arrows- which is attributed to alignment and elongation in the flow. The authors should clarify and better depict how the β -sheet peptidic chain axis of the assembled beads (red lines) are oriented as I cannot deduce this from the comparison of the oligomer with the short-rod like SNFs. Are they parallel to the flow direction or normal to the flow direction? If the model would apply to biospinning one would expect alignment along the rod axis, in the same way as β -sheets are aligned along the silk fiber axis.

Reply: We appreciate the reviewer's insightful question regarding the β -sheet peptide chain orientation in Figure 1a. We agree that the assembled oligomers likely exhibit a preferred orientation of their β -sheet peptide chain axis, potentially aligned parallel to the elongation axis of the silk nanofibrils. This alignment is supported by previous studies on silk spinning, which suggest that during the process, antiparallel β -sheet peptide chains tend to align parallel to the elongation axis of the fiber. This alignment is believed to optimize packing efficiency and facilitate strong intermolecular interactions (Nat. Commun, 2022, 13, 4329; J. Mater. Chem. B, 2016, 4, 4337). Additionally, research has shown that a higher degree of β -sheet crystallite orientation along the fiber axis often translates to a higher breaking stress (Small, 2019, 15, 1805294; Soft Matter, 2014, 10, 2116).

While directly verifying the β -sheet peptide axis orientation within our static experimental system remains challenging, our simulation results (Figure 4c and Figure S19) do suggest a preferred orientation – parallel to the elongation axis of the silk nanofibrils. We have incorporated more detailed discussions regarding the β -sheet orientation in the revised manuscript.

Changes made in the revised manuscript:

We thank the reviewer for this question, in response to which we have made the following changes in Figure 1a, Lines 285-288 on page 10, Lines 312-313 on Page 11 in the revised manuscript:

“

Fig. 1: SNF assembly and graphene plasmon-enhanced FTIR. **a** Schematic diagram of speculative SNF assembly mechanism: Unfolded SF molecules first undergo a conformational change, forming β -sheet-rich oligomers; these oligomers then connect with nearby molecules and elongate into short-rod-like SNFs, with the β -sheet chains aligned along the elongation direction (indicated by grey dashed arrow). Finally, these short SNFs further align and elongate, becoming thicker and longer to form long-rod-like SNFs. **b** Schematic diagram of graphene plasmon-enhanced FTIR for measuring SNF assembly (left panel) and a typical AFM of SNF adsorbed on the graphene nanoribbons (right panel). **c** The extinction spectra of graphene plasmon with (red curve) and without SNFs (black dashed curve). T_{V_G} is measured when V_G is -100 V, while T_0 is measured when V_G is 100 V. The black curve is extinction spectrum of bulk SNF film which is reduced by 10 times for comparing. The assembly process occurs in an aqueous solution at 348 K, with a duration of 300 min.”

Lines 281-283 on page 10 in revised manuscript: “Moreover, as shown in Fig. 4c and Fig. S19, the β -sheet peptide chain axis of the assembled oligomers tends to be parallel with the elongation axis of SNF at T , which is consistent with previous reports^{64, 65}.”

Lines 312-313 on Page 11 in revised manuscript: “A SNF assembly model is thus provided as illustrated in Fig. 1a. In nucleation stage, unfolded SF molecules undergo a conformational transition to form β -sheet-enriched oligomers. These oligomers then progress to the growth stage, connecting with adjacent molecules and elongating into short-rod-like SNFs. The β -sheet peptide chain axis of the assembled oligomers has a preferred orientation – parallel to the elongation axis of the SNFs.”

Reviewer #2 (Remarks to the Author):

I am pleased to confirm that I am satisfied with the changes made to the manuscript in its revised form. All my concerns have been properly addressed. As a result, I am happy to recommend this for publication.

Reply: We appreciate the reviewer's recommendation for publication.

Reviewer #3 (Remarks to the Author):

I appreciate the revised manuscript and the response from the authors to the reviewer's comments. Many of them are resolved, but some concerns persist.

The numbering here refers to my previous review comments.

2-1. Graphene plasmon-enhanced FTIR: While the reviewer appreciates the response from the authors, the reviewer would suggest that the authors add some of those details to their manuscript. The reviewer overall feels that the important details of characterization conditions are still missing.

Reply: We appreciate the reviewer for the kind suggestions. We have incorporated the suggestions by adding detailed discussion and supplementary characterization data (see details below).

a) The graphene pattern design is very important to “observe” silk nanofibril assembly in the reported work, as the pattern design is directly related to the plasmonic properties of the “sensor.” The reviewer still thinks sufficient experimental details are not provided. Specifically, what exactly was the width of the graphene ribbons? The authors mentioned that “we designed graphene nanoribbons with width of 50-90 nm,” but it is not sufficient. What was the exact width the authors aimed for? As the authors mentioned the resolution of the E-beam lithography was 5 nm, the width range of 40 nm is not reasonable. In addition, the reviewer recommends that the authors add information about the width of graphene ribbons and the period-to-width ratio in the manuscript.

Reply: We appreciate the reviewer's inquiry regarding the graphene nanoribbon (GNR) dimensions. As clarified in the revised Figure S3, this work utilizes GNRs with a width of approximately 60 nm and a period-to-width ratio of 2:1. Details about the specific dimensions for different sensors can now be found within the Methods section.

It's important to clarify that the observed range in GNR widths (~50-90 nm) is not a result of errors during electron beam lithography (EBL). Instead, this variation reflects the intentional design of different nanoribbon sizes. Notably, wider GNRs require a higher graphene Fermi energy (E_F), which is proportional to the change in relative gate voltage ΔV , to achieve the same resonance frequency for the graphene plasmon (Nat. Photon., 2013, 7, 394; ACS Nano, 2014, 8, 1086). However, applying a high ΔV_G carries the risk of device breakdown, potentially compromising subsequent measurements (ACS Photonics 2014, 1, 3, 135).

Therefore, for this study, we strategically selected GNR widths of 55 nm, 60 nm, and 65 nm to balance these considerations. The design strategy for these GNRs is elaborated on in Supplementary Note 1.

b) Please add the discussions related to the graphene design process (as the authors provided in the response) in the supplementary information or manuscript.

Reply: We are grateful to the reviewer for raising the suggestion. the discussions related to the graphene design process is supplemented in main content and Supplementary Note 1 of revised version.

Changes made in the revised manuscript:

We thank the reviewer for this question, in response to which we have supplemented new Figure S3 and rewrite sentences in Lines 100-107 of revised manuscript:

New Figure S3 is supplemented in revised Supplemental information:

“

Supplementary Figure S3. Scanning electron microscope images of graphene nanoribbons.

(a) Device 1 as utilized in Figure 1c. The graphene nanoribbon width is 55 nm, and period is 110 nm. (b) Device 2 as utilized in Figure 2a&3b. The graphene nanoribbon width is 60 nm, and period is 120 nm. (c) Device 3 as utilized in Figure 2b&3c. The graphene nanoribbon width is 60 nm, and period is 120 nm. (d) Device 4 as utilized in Figure 2c&3d. The graphene nanoribbon width is 65 nm, and period is 120 nm.

”

Lines 100-107 on Page 4 of the revised manuscript:

“To probe the assembly process of SNFs by FTIR, we utilized a graphene plasmonic infrared sensor (details in Fig. S2). The graphene nanoribbon width is ~60 nm and period-to-

width ratio $\approx 2:1$ as shown in Fig. S3. The sensor was designed to a field-effect transistor based on monolayer graphene nanoribbons, as shown in Fig. 1b and Fig. S4. The gate voltage (V_G) was also applied for dynamically tuning the resonance frequency of graphene plasmon to target molecular fingerprints (details in Note 1 and Fig. S5 of Supplementary information)^{42, 43.}”

Lines 34-60 on Page 3 of the revised supplementary information:

“**Supplementary Note 1:** The dispersion of graphene plasmon is^{1, 2:}

$$\omega_{pl} = \sqrt{\frac{e^2 |E_F| q}{2\pi \hbar^2 \epsilon_0 \epsilon_r}},$$

where ϵ_0 is the dielectric constant of air, ϵ_r is the average dielectric constant of its surrounding medium, E_F is the fermi energy of graphene, q is the wave vector and $q = \pi/W$ (W is the width of graphene nanoribbon). Thus, the resonance frequency of graphene plasmon (ω_{pl}) is related to W and E_F of graphene nanoribbon. In addition, when the width of graphene nanoribbon is close to the spacing between them, the energy loss due to plasmon-plasmon interactions between adjacent ribbons is minimized. This optimizes the extinction of graphene plasmons, a crucial factor for our sensor's performance³. To achieve the highest sensitivity for SNF assembly detection, we carefully designed the W and E_F (proportional to ΔV_G , $\Delta V_G = |V_G - V_{CNP}|$) in a cooperative manner. A larger W requires a higher ΔV_G to reach the desired resonance frequency for optimal overlap with the fingerprint region of target molecules⁴. However, applying a high ΔV_G carries the risk of device breakdown, potentially compromising subsequent measurements⁵. Therefore, we strategically chose W of approximately 60 nm and period-to-width ratio of 2:1 to balance these considerations.

As shown in Figure S5b, d, and f, the initial doping level of graphene can vary between sensors due to fabrication processes, leading to different charge neutral points (V_{CNP}). To address this, we performed $I_{SD}-V_G$ characterization to determine the V_{CNP} , and measured the background FTIR transmittance (T_0) at V_{CNP} . We then get Extinction spectrum (Extinction = $1 - T_{V_G}/T_0$, where T_{V_G} represents the transmittance measured at a specific gate voltage (V_G)) by tuning the gate voltage (V_G) to achieve the desired resonance frequency that overlaps with the Amide I band. Figure S5i exemplifies this process, showing the dynamic response of graphene plasmon from 1450-1750 cm^{-1} as ΔV_G is swept from 59 V to 119 V. Ultimately, we select the appropriate gate voltage ($\Delta V_G = 84$ V, red curve) for subsequent FTIR measurements and analysis, where the graphene plasmon response best aligns with the target molecule's fingerprint region.”

c) In the revised manuscript (LL112-118), the authors mentioned that “the graphene nanoribbon widths were designed and VG is tuned to excite...” So, how high (or low) was the gate voltage to excite graphene plasmon in the nanoribbon? Did you use the same gate voltage for all measurements?

Reply: As mentioned in the figure captions of Figures S9, S11, and S13, different gate voltages are indeed used for FTIR measurements to make sure the ΔV_G are the same, which

determined the graphene Fermi level in this equation $\omega_{pl} = \sqrt{\frac{e^2|E_F|q}{2\pi\hbar^2\varepsilon_0\varepsilon_r}}$. The detailed values are listed in the figure captions. Figure S5i exemplifies this process, showing the dynamic response of graphene plasmon from 1450-1750 cm^{-1} as ΔV_G is swept from 59 V to 119 V. By strategically adjusting ΔV_G , we can tune the resonance frequency of the graphene plasmon to achieve optimal overlap with the target molecule's fingerprint region in the Amide I band. Ultimately, we select the most suitable gate voltage ($\Delta V_G = 89$ V, blue curve) for subsequent FTIR measurements and analysis.

Changes made in the revised manuscript:

We thank the reviewer for this question, in response to which we have made the following changes:

Lines 34-60 on Page 3 of the revised supplementary information:

“Supplementary Note 1: The dispersion of graphene plasmon is^{1, 2}:

$$\omega_{pl} = \sqrt{\frac{e^2|E_F|q}{2\pi\hbar^2\varepsilon_0\varepsilon_r}},$$

where ε_0 is the dielectric constant of air, ε_r is the average dielectric constant of its surrounding medium, E_F is the fermi energy of graphene, q is the wave vector and $q = \pi/W$ (W is the width of graphene nanoribbon). Thus, the resonance frequency of graphene plasmon (ω_{pl}) is related to W and E_F of graphene nanoribbon. In addition, when the width of graphene nanoribbon is close to the spacing between them, the energy loss due to plasmon-plasmon interactions between adjacent ribbons is minimized. This optimizes the extinction of graphene plasmons, a crucial factor for our sensor's performance³. To achieve the highest sensitivity for SNF assembly detection, we carefully designed the W and E_F (proportional to ΔV_G , $\Delta V_G = |V_G - V_{\text{CNP}}|$) in a cooperative manner. A larger W requires a higher ΔV_G to reach the desired resonance frequency for optimal overlap with the fingerprint region of target molecules⁴. However, applying a high ΔV_G carries the risk of device breakdown, potentially compromising subsequent measurements⁵. Therefore, we strategically chose W of approximately 60 nm and period-to-width ratio of 2:1 to balance these considerations.

As shown in Figure S5b, d, and f, the initial doping level of graphene can vary between sensors due to fabrication processes, leading to different charge neutral points (V_{CNP}). To address this, we performed $I_{SD}-V_G$ characterization to determine the V_{CNP} , and measured the background FTIR transmittance (T_0) at V_{CNP} . We then get Extinction spectrum (Extinction = $1 - T_{V_G}/T_0$, where T_{V_G} represents the transmittance measured at a specific gate voltage (V_G)) by tuning the gate voltage (V_G) to achieve the desired resonance frequency that overlaps with the Amide I band. Figure S5i exemplifies this process, showing the dynamic response of graphene plasmon from 1450-1750 cm^{-1} as ΔV_G is swept from 59 V to 119 V. Ultimately, we select the appropriate gate voltage ($\Delta V_G = 84$ V, red curve) for subsequent FTIR measurements and analysis, where the graphene plasmon response best aligns with the target molecule's fingerprint region.”

, and addition of Figure S5i in revised supplementary material:

Supplementary Figure S5. Electrical performance of graphene plasmonic infrared sensors. (a) The I_{SD} - V_{SD} data, and (b) I_{SD} - V_G data of the graphene plasmonic sensor (Device 1), as utilized in Fig. 1c. (c) The I_{SD} - V_{SD} data, and (d) I_{SD} - V_G data of the graphene plasmonic sensor (Device 2), as utilized in Fig. 2a&Fig. 3b. (e) The I_{SD} - V_{SD} data, and (f) I_{SD} - V_G data of the graphene plasmonic sensor (Device 3), as utilized in Fig. 2b&Fig. 3c. (g) The I_{SD} - V_{SD} data, and (h) I_{SD} - V_G data of the graphene plasmonic sensor (Device 4), as utilized in Fig. 2c&Fig. 3d. (i) **The tunable graphene plasmon when varying gate voltage (V_G) relative to charge neutrality point (V_{CNP}) ($\Delta V_G = V_G - V_{CNP}$) measured by Device 4. V_{SD} and I_{SD} denote the applied voltage and the measured current between the source and drain, respectively. V_{SD} - I_{SD} curves were recorded with V_G set to 0 V, while V_G - I_{SD} curves were obtained with V_{SD} at 0.1 V. The linear slope observed in the V_{SD} - I_{SD} curve suggests good contact between the graphene and the electrodes. The lowest point on the V_G - I_{SD} curve indicates the V_{CNP} . As shown in Fig. S4h, applying a V_G lower than the V_{CNP} (indicated by the red shaded area) shifts the Fermi level towards the valence band, indicating p-doping of graphene.”**

2-2. Materials

a) It is difficult to agree that the graphene used in this work is monolayer while the reviewer

understands the authors' claim on the G' peak profile. First, the intensity ratio ($I_{G'}/I_G$) is clearly less than 2. Second, the AFM profile in Figure S3 does not reflect the claim. It is very common that the edge of patterned graphene shows a spike in AFM height profile, but the height difference between the center of the pattern and the substrate is clearly >1 nm, which implies that the pattern has 2 or more layers of graphene.

Reply: In this work, we confirm that the utilized graphene is indeed a monolayer. The Raman spectrum of graphene reveals an $I(G')/I(G)$ ratio of approximately 1.8, slightly less than 2. This can be attributed to the p-type doping effect which can be implied from measured $I_{SD}-V_G$ curve in Figure S5. In graphene, the doping has two major consequences: a change of the equilibrium lattice parameter with a consequent stiffening/softening of the phonons, and (2) the onset of effects beyond the adiabatic Born–Oppenheimer approximation that modify the phonon dispersion close to the Kohn anomalies (Nature Nanotech., 2008, 3, 210; Phys. Rev. B 2009, 80, 233407). As the doping concentration increases in graphene, the $I(G')/I(G)$ ratio typically decreases and may fall below 2, as illustrated in Figure R1a.

A more reliable approach to determining the graphene layer number involves analyzing the number of contributing peaks within the G' band, as shown in Figure R1b (Ref. Phys. Rev. Lett., 2006, 97, 187401; Nano Research 2008, 1, 273). When the G' peak comprises a single Gaussian peak, it signifies a monolayer graphene structure. Figure R1c presents the fitted Raman spectrum of our graphene. As evident from the figure, the G' peak consists of a single Gaussian peak, confirming that the utilized graphene is indeed a single layer.

Figure R1 (a) The ratio of the intensity of the 2D(G') peak in the Raman spectrum to the intensity of the G peak exhibits a clear dependence on the electron concentration. Citation: Nature Nanotechnology 2008, 3(4): 210-215. (b) The enlarged 2D(G') band Raman spectra of graphene (1, 2, 3, and 4 layers) with curve fitting. Citation: Nano Research 2008, 1(4): 273-291. (c) The Raman spectrum of graphene utilized in this work, and the enlarged G' band with single Gauss peak fitting.

Atomic force microscopy (AFM) measurements of our monolayer graphene revealed a height exceeding the theoretical value of 0.34 nm. This discrepancy can be attributed to residual poly(methyl methacrylate) (PMMA) on the graphene surface. PMMA is employed as an electron beam resist during the fabrication of our graphene plasmonic devices, as detailed in

the Methods section. However, completely removing these PMMA residues poses a significant challenge, often leaving behind layers 1-2 nm thick (ACS Nano 2011, 5, 2362). The presence of these residual PMMA layers contributes to the increased height measured by AFM.

Changes made in the revised manuscript:

We thank the reviewer for these questions, in response to which we have made the following changes include Lines 111-113 in Page 4, Figure S2 and Figure S4 of the revised manuscript and supplementary information:

Lines 111-113 in Page 4 of the revised manuscript: “The observed height of the monolayer graphene exceeding 0.34 nm is attributed to poly(methyl methacrylate) (PMMA) residue contamination from the sensor fabrication⁴⁴.”

Figure S2 of the revised supplementary information:

“

Supplementary Figure S2 Characterization of graphene plasmonic infrared sensor. (a) Optical microscope image of graphene film and graphene nanoribbon region, which shows the graphene is continuous. (b) Raman spectrum of utilized graphene film. 514 nm laser is used. By fitting the Raman spectrum of graphene we measured, as magnified in the left box, the G' peak is observed to be a single peak, confirming that it is single layer graphene¹⁵⁻¹⁷.

”

Figure S4 of the revised supplementary information:

“
Supplementary Figure S4. The AFM data of graphene nanoribbons with and without SNFs. The morphology and extracted height of bare graphene nanoribbons. **Although the theoretical thickness of a monolayer of graphene is 0.34 nm, our AFM measurements indicate that the height of the monolayer graphene exceeds 1 nm. This discrepancy is attributed to the use of poly(methyl methacrylate) (PMMA) during the fabrication of graphene plasmonic devices¹⁸, which results in 1-2 nm thick PMMA residues on the graphene surface. These residues are challenging to completely eliminate^{19,20}.** (b) The morphology and extracted height of graphene nanoribbons with SNFs.
 ”

b) Optical microscopy image (Figure S2) could support the continuity and monolayer. However, the pattern here is too fine to show anything with an optical microscope. The reviewer was curious why the authors did not use scanning electron microscopy to show the patterns of the graphene layer.

Reply: In response to the reviewer's valuable suggestion, we have incorporated scanning electron microscopy (SEM) data of the utilized graphene nanoribbons within the revised supplementary information. These images will provide valuable insights into the morphology and dimensions of the nanoribbons employed in our experiments.

Changes made in the revised manuscript:

We thank the reviewer for these questions, in response to which we have added Figure S3 in the revised supplementary information:

“

Supplementary Figure S3. Scanning electron microscope images of graphene nanoribbons. (a) Device 1 as utilized in Figure 1c. The graphene nanoribbon width is 55 nm, and period is 110 nm. (b) Device 2 as utilized in Figure 2a&3b. The graphene nanoribbon width is 60 nm, and period is 120 nm. (c) Device 3 as utilized in Figure 2b&3c. The graphene nanoribbon width is 60 nm, and period is 120 nm. (d) Device 4 as utilized in Figure 2c&3d. The graphene nanoribbon width is 65 nm, and period is 120 nm.

”

Based on the revised manuscript, comments from all reviewers, and response to the review comments, I am not fully convinced that this manuscript is acceptable for publication in Nature Communications, considering the broad readership of the journal. I clearly see new findings and the reported work's contributions to the silk nanofibril research communities, so perhaps, it is more suitable to publish this manuscript in a more specialized journal.

Reply: We are grateful for the reviewers' careful review of our manuscript, particularly their recognition of the novel findings and contributions to silk nanofibril research. We have meticulously addressed all comments and suggestions provided, resulting in significant improvements to the quality and clarity of the manuscript. We are confident that our revised work offers valuable insights and merits consideration for publication in Nature Communications.

REVIEWERS' COMMENTS

Reviewer #1 (Remarks to the Author):

I agree to the modifications to the revised text. In particular, the argument on a beta-sheet fraction, which cannot be detected by XRD, is correct. A fraction of very small beta-sheet nuclei has already been proposed by Simmons et al., (*Science* (1996), 271, 84-87), based on NMR. Such nuclei would appear in XRD experiments only in the diffuse scattering background. I agree also to changes to Figure 1.

RESPONSE TO REVIEWERS' COMMENTS

We thank all referees for their careful reviewing of this work and the recommendation for the publication of this work. Please, find our point-by-point reply below.

Reviewer #1 (Remarks to the Author):

I agree to the modifications to the revised text. In particular, the argument on a beta-sheet fraction, which cannot be detected by XRD, is correct. A fraction of very small beta-sheet nuclei has already been proposed by Simmons et al., (Science (1996), 271, 84-87), based on NMR. Such nuclei would appear in XRD experiments only in the diffuse scattering background. I agree also to changes to Figure 1.

Reply: We thank the reviewer for the thoughtful review and agreement with the modifications made to the revised text and Figure 1. Your feedback has been valuable in enhancing the clarity and accuracy of our manuscript.